# 3D Crustal Density Model of the Marmara Sea

Ershad Gholamrezaie[1,2], Magdalena Scheck-Wenderoth[1,3], Judith Sippel[1], Oliver Heidbach[1], and Manfred R. Strecker[2]

[1]Helmholtz Centre Potsdam–GFZ German Research Centre for Geosciences, Potsdam, Germany
[2]Institute of Earth and Environmental Science, University of Potsdam, Germany
[3]Faculty of Georesources and Material Engineering, RWTH Aachen, Aachen, Germany

*Correspondence to*: Ershad Gholamrezaie (ershad@gfz-potsdam.de)

**Abstract.** The Sea of Marmara, in Northwest Turkey, is a transition zone where the dextral North Anatolian Fault Zone (NAFZ) propagates westward from the Anatolian plate to the Aegean plate. The area is of interest in the context of seismic
hazard of Istanbul, a metropolitan area with about 15 million inhabitants. Geophysical observations indicate that the crust is heterogeneous beneath the Marmara Basin, but a detailed characterization of the crustal heterogeneities is still missing. To assess if and how crustal heterogeneities are related to the NAFZ segmentation below the Marmara Sea, we develop new crustal-scale 3D density models which integrate geological and seismological data and that are additionally constrained by 3D gravity modelling. For the latter, we use two different gravity datasets including global satellite data and local marine
gravity observation. Considering the two different datasets and the general non-uniqueness in potential field modelling, we suggest three possible "endmember" solutions that are all consistent with the observed gravity field and illustrate the spectrum of possible solutions. These models indicate that the observed gravitational anomalies originate from significant density heterogeneities within the crust. Two layers of sediments, one syn-kinematic and one pre-kinematic with respect to the Marmara Sea formation are underlain by a heterogeneous crystalline crust. A felsic upper crystalline crust (average
density of 2720 kg.m$^{-3}$) and an intermediate to mafic lower crystalline crust (average density of 2890 kg.m$^{-3}$) appear to be crosscut by two large, dome-shaped mafic high-density bodies (density of 2890 to 3150 kg.m$^{-3}$) of considerable thickness above a rather uniform lithospheric mantle (3300 kg.m$^{-3}$). The spatial correlation between two major bends of the main Marmara fault and the location of the high-density bodies suggests that the distribution of lithological heterogeneities within the crust controls the rheological behaviour along the NAFZ, and consequently, maybe influences fault segmentation and
thus the seismic hazard assessment in the region.

## 1. Introduction

The Sea of Marmara in NW Turkey is an extensional basin associated with a right-stepping jog in the orientation of the North Anatolian Fault Zone (NAFZ; Fig. 1), a westward-propagating right-lateral strike-slip fault that constitutes the plate boundary between the Anatolian and the Eurasian plates (Fig. 1a; McKenzie, 1972; Şengör et al., 2005). As one of the most
active plate-bounding strike-slip faults in the world, and being located in the Istanbul metropolitan area with a population of

approximately 15 million, the NAFZ has been the focus of numerous geoscientific investigations over the past decades (e.g. Barka, 1996; Ambraseys, 1970; Stein et al., 1997; Armijo et al., 1999; Şengör et al., 2005; Le Pichon et al., 2015). Several recent research programs (e.g. SEISMARMARA: Hirn and Singh, 2001: http://dx.doi.org/10.17600/1080050; GONAF: Bohnhoff et al., 2017a: https://www.gonaf-network.org; MARsite: http://www.marsite.eu) have been embarked on to improve the observational basis for the seismic hazard assessment in the Marmara Sea region.

The Marmara section of the NAFZ, the Main Marmara Fault (MMF; Le Pichon et al., 2001; 2003), is considered to be a 150-km-long seismic gap between the ruptures of two strong events in 1912 (M 7.3) and 1999a (M 7.4), is a zone of strong earthquakes (M ~ 7.4) with a recurrence time of approximately 250 years (Fig. 1b and 1c); this section experienced the last earthquake in 1509 and 1766, suggesting that the fault is mature and that the potential for a large seismic event is regarded is high (Ambraseys, 2002; Barka et al., 2002; Parsons, 2004; Janssen et al. 2009; Murru et al., 2016; Bohnhoff et al., 2013; 2016a; 2016b; 2017b). A key question is if this 150-km-long seismic gap will rupture in the future in one event or in several separate events due to segmentation of the MMF, an issue that will depend a lot on the stress evolution along strike among other forcing factors. In this regard, three-dimensional (3D) geological models are the fundament of geomechanical models, and the distribution of density is of key importance as density controls body forces. Density modelling is generally done by integrating geological information, seismic observations, and gravity data. Furthermore, gravity models can also help to assess the density distribution at greater depth where borehole observations and/or seismic surveys have limitations.

Our study aims to evaluate the deep crustal configuration of the Marmara Sea and surrounding areas. To address the question if there is a spatial relationship between fault activity and the distribution of certain physical properties in the crust, we develop 3D density models that integrate available seismological observations and are consistent with observed gravity measurements. In a previous gravity modelling effort (Kende et al., 2017), an inversion method was applied to calculate the Moho depth below the Marmara region. Building on an earlier 3D structural model developed to evaluate the stress-strain state in this region (Hergert and Heidbach, 2010; 2011; Hergert et al., 2011), we use crustal and regional-scale forward 3D gravity modelling and seismic data as additional constraints. In addition, we compare and discuss our results with the previously published results of Kende et al. (2017). This comparison confirms that significant density heterogeneities are laterally present within the crust below the Sea of Marmara. In particular, we find indications for lateral density heterogeneities within the crust in the form of two local high-density bodies that may influence the kinematics of the NAFZ below the Sea of Marmara.

## 1.1. Geological setting

In the large-scale plate-tectonic framework of Asia Minor, the NAFZ accommodates the westward escape of the Anatolian plate in response to the northward motion and indentation of the Arabian plate into Eurasia and westward enlarging of the deep slab detachment beneath the Bitlis–Hellenic subduction zone (Fig. 1a: McKenzie, 1972; Şengör et al., 2005; Faccenna et al., 2006; Jolivet et al., 2013); this has resulted in numerous deformation features along the well-defined trace of the fault

and regionally, along the northern flanks of the Anatolian Plateau (Barka and Hancock, 1984; Barka, and Reilinger, 1997; Pucci et al., 2006; Yildirim et al. 2011; 2013).

In the westernmost sector, the NAFZ bifurcates into several strands of locally variable strikes, which has resulted in a mosaic of pull-apart basins flanked by steep mountain fronts and intervening structural highs. All of these morphotectonic features
of the greater Marmara region are characterized by active Quaternary deformation process (Yildirim and Tüysüz, 2017). To the west of the Almacik Block, a transpressional push-up ridge, the NAFZ splits into three main strands (Fig. 1b; Armijo et al., 1999, 2002): the northern, middle, and southern branches. The northern branch traverses the Sea of Marmara and forms the N70°E striking Main Marmara Fault (MMF; Le Pichon et al., 2001; 2003). The approximately E-W-striking middle branch passes through the Armutlu Peninsula and continues along the southern coast of the Marmara Sea; this branch
changes strike to NE-SW in the southern part of the Kapıdağ Peninsula (Yaltırak and Alpar, 2002; Kurtuluş and Canbay, 2007). The southern branch traverses the Biga Peninsula, the region to the south of the southern margin of the Marmara Sea. The Marmara Sea is an E-W elongated transtensional basin with up to 1300 m water depth along its axial part; it is surrounded by onshore domains at about 600 m average elevation (Fig. 1c). The deepest part of the basin is the North Marmara Trough (NMT: Laigle et al., 2008; Bécel et al., 2009), which hosts three main sedimentary basins along the NAFZ.
These include the Çınarcık, Central, and Tekirdağ basins. These depocentres are separated from each other by the shallower Central High (East) and the Western High (West), respectively. In the deep parts of the basin protracted subsidence has resulted in the accumulation of more than 5 km of Pliocene–Holocene sediments (Le Pichon et al. 2001; 2003; 2015; Armijo et al. 2002; Parke et al. 2002; Carton et al., 2007; Laigle et al., 2008; Bécel et al. 2009; 2010; Bayrakci et al., 2013).

The region of the Marmara Sea is an integral part of the NAFZ, which has begun its activity in the east approximately 13 to
11 Ma ago (Şengör et al., 2005). Although different models and timing constraints for the onset of basin formation in the Marmara Sea have been presented in the context of the evolution of the NAFZ and the Aegean region (e.g., Armijo et al., 1999; Ünay et al., 2001; Yaltırak, 2002; Şengör et al., 2005; Le Pichon et al., 2014; 2015), offset geological marker horizons, displaced structures, and paleontological data point to a transtensional origin during the propagation and sustained movement of the NAFZ with displacement and block rotations after the Zanclean transgression in the early Pliocene. Such a
geodynamic scenario of transtensional dextral strike-slip faulting is compatible with space-geodetic data, the pattern of seismicity and geomorphic indicators in the landscape (Reilinger et al., 1997; 2006; Barka and Kadinsky-Cade, 1988; Bürgmann et al., 2002; Pucci et al., 2006; Akbayram et al., 2016; Yildirim and Tüysüz, 2017).

In contrast, based on GPS velocity data and surface geological observations, there are also arguments that the kinematics of the MMF correspond to a pure right-lateral strike-slip with the exception of the Çınarcık Basin area that the bend of the
Princes Islands segment causes a transtensional setting (e.g. Le Pichon et al., 2003; 2015).

## 2. Method and model setup

Like for the earlier 3D model (Hergert and Heidbach, 2010), our study area extends from 40.25° N–27.25° E to 41.15° N–30.20° E and is projected as a rectangular shape in WGS84 UTM Zone 35N with a dimension of 250-by-100 km (Fig. 1c). It covers the Sea of Marmara and the adjacent onshore areas, as well as the city of Istanbul and the Bosporus.

The principal approach used for this study is crustal-scale 3D gravity forward modelling to assess the density configuration of different structural units. In this methodology, the gravity response of a model is calculated and compared with the observed gravity field. The model is iteratively modified to find the best-fit with observations. Since the solution is not unique in gravity modelling, it is required to reduce the number of free parameters by integrating other available geophysical and/or geological data as additional constraints. In the spirit of this philosophy, the workflow adopted in this study consists

in: (1) setting up an initial density model (Fig. 2 and 3) – in our case based on the previous studies (Hergert and Heidbach, 2010; 2011; Hergert et al., 2011); (2) calculating the gravity response of this initial model and analysing the misfit (gravity residual) between modelled and observed gravity; (3) modifying the initial model by introducing additional density variations while integrating additional constraining data to obtain the density–geometry configuration that reproduces the observed gravity field best. In general, positive residual anomalies indicate that more mass is required in the model to fit the

observed gravity field, whereas negative residuals imply that the mass in the model is too large in the domain of the misfit.

3D forward gravity modelling has been performed using the Interactive Gravity and Magnetic Application System–IGMAS+ (Transinsight GmbH©; Schmidt et al, 2011). In IGMAS+, the gravity response of a 3D structural and density model is calculated and compared with the observed gravity field over the model area. Therefore, the model has to be defined in terms of the geometric configuration of its individual structural units. In addition to geometry, information on the densities needs to

be assigned to the different units of the model to calculate the gravity response. The chosen parameter combinations for the different models studied are detailed in Sect. 4. IGMAS+ provides the density-geometry configuration in the form of triangulated polyhedrons over the 3D model domain. These polyhedrons are spanned between 2D vertical working sections where the model can be interactively modified (Schmidt et al., 2011). For this study, a lateral resolution of 2500 m is considered that results in 100 North-South oriented working sections. Downward the models extend to a constant depth of 50

km b.s.l. and the unit comprised between the Moho and the lower model boundary is considered as the uniform lithospheric mantle. To avoid lateral boundary effects, the models extend on all sides 370 km further than the study area.

Key horizons where major contrasts in density are expected are the air-water interface, the sediment-water interface, the interface separating sediments and crystalline crust and the crust-mantle boundary (Moho). These interfaces also are well imaged with seismic methods and can therefore easily be integrated. Internal heterogeneities within the crust, may not be

identified by seismic methods or only locally along individual profiles. This is where 3D gravity modelling can be used in addition to translate velocities to densities first along the seismic section and use density modelling to close the gaps in between. This strategy together with the three-dimensionality of the calculation strongly reduces model uncertainties imposed by the general non-uniqueness of gravity modelling as densities need to be in certain ranges for different rock types

and density anomalies at different depths produce gravity effects of different wavelengths (e.g. Schmidt et al., 2011; Maystrenko et al., 2013; Sippel et al., 2013; Maystrenko and Scheck-Wenderoth, 2013).

To assess the density variations in the deeper crust of the Marmara Sea region, we calculate the gravity response for models of increasing complexity concerning their 3D structural and density configuration: (1) the initial model with homogeneous crust below the sediments, (2) a more differentiated model integrating additional seismic observations for the different crustal levels below the sediments, and (3) a series of final best-fit models in which the remaining residual anomaly is minimized by implementing additional density-geometry changes in the crust but respecting the seismic data. As two different gravity datasets are available, we calculate the difference between model response and observed gravity for both datasets.

Throughout the modelling procedure, the uppermost surface, the bathymetry (Fig. 1c), the top-basement depth (Fig. 2a) and the depth to the Moho discontinuity (Fig. 2b) are kept fixed as defined in the initial model since the geometries of these interfaces are well-constrained by geological and geophysical data. In all tested models, an average density of 1025 kg.m$^{-3}$ was assumed for seawater, and a homogeneous density of 3300 kg.m$^{-3}$ is assigned to the mantle below the Moho. For all gravity models presented, we define the uppermost surface of the model as the onshore topography and as the sea level offshore. Accordingly, the thickness between sea level and bathymetry (Fig. 1c) corresponds to the column of seawater (Fig. 3a) which attains the largest values in the Tekirdağ, Central, and Çınarcık basins.

## 3. Input data

The database for this study includes topography-bathymetry data, geometrical and density information from a previous 3D structural model, seismic observations, and different sets of published free-air gravity data including shipboard gravity dataset.

### 3.1. Topography and bathymetry

The topography–bathymetry (Fig. 1c) was exported from 1 Arc-Minute Global Relief Model (ETOPO1; Amante and Eakins, 2009). This dataset, over the study area, integrates the 30 arc-second grid obtained from NASA's Shuttle Radar Topography Mission (SRTM) and a bathymetry dataset (MediMap Group, 2005) with 1 km resolution. In addition, to increase the bathymetry resolution within the North Marmara Trough, high-resolution multibeam (EM300) acquired bathymetry (Le Pichon et al., 2001) is integrated into the model (Fig. S1 in the Supplement).

Figure 1c illustrates that the present-day Marmara Sea is surrounded by up to 1500 m high regions. The configuration of the present-day sea floor shows that the Marmara Sea is structured into the three main depocentres of the Tekirdağ Basin, the Central Basin, and the Çınarcık Basin where the water depth reaches up to 1300 m. While the axis of the Central Basin is aligned along the MMF, the Çınarcık Basin and the Tekirdağ Basin extend only south and mostly north of the MMF, respectively. The MMF bends along the northern boundary of the Çınarcık Basin, at the Tuzla Bend, from an E-W directed

strike (East of the Marmara Sea) to an ESE-WNW strike direction at the north-western margin of the Çınarcık Basin before it resumes the E-W strike direction at the Istanbul Bend. The segment of the MMF between the two bends is the Princes Islands Segment. Farther in the West of the Marmara Sea, at the Ganos Bend, the MMF once more changes strike direction from E-W to ENE-WSW. There, the MMF exits the Sea of Marmara and creates the Ganos Fault segment of the NAFZ.

### 3.2. Initial model

The 3D structural model (Fig. 3: Hergert and Heidbach, 2010; Hergert et al., 2011), considered to be the initial model for our study, differentiates three main horizons: (1) the topography–bathymetry (Fig. 1c), (2) a top-basement surface (Fig. 2a), and (3) the Moho discontinuity (Fig. 2b). In their study, Hergert and Heidbach (2010), modeled the top-basement geometry based on seismic observations (Parke et al., 2002; Carton et al., 2007; Laigle et al., 2008; Bécel et al., 2009; 2010) and other geophysical and geological data such as 3D seismic tomography (Bayrakci, 2009), well data (Ergün and Özel, 1995; Elmas, 2003) and geological maps (Elmas and Yigitbas, 2001). This surface, however, has been interpreted by others as the top of a Cretaceous limestone that is pre-kinematic with respect to the opening of the Sea of Marmara (Ergün and Özel, 1995; Parke et al., 2002; Le Pichon et al., 2014). Hergert and Heidbach (2010) derived the thickness of the sediments of the Marmara Sea as the difference between bathymetry-topography and top-basement. Accordingly, their "basement" delineates the base of the sediments and not the crystalline basement. First deep seismic surveys in the Sea of Marmara (Fig. 4: Laigle et al., 2008; Bécel et al., 2009) indicate that this basement is a pre-kinematic basement with respect to the opening of the Marmara Sea. Accordingly, Laigle et al. (2008), suggests the term of "syn-kinematic" infill for the sediments above the pre-kinematic basement. We, therefore, regard these sediments as the "syn-kinematic sediments" and refer to top-basement of the initial model as the "base syn-kinematic sediments" in the following.

The syn-kinematic sediments in our model represent the deposits related to the opening of the Marmara Sea and are interpreted to be mainly Pliocene–Quaternary infill (Laigle et al., 2008; Bécel et al., 2010; Bayrakci et al., 2013; Le Pichon et al., 2015). Accordingly, they are mostly missing in the domains outside the Marmara Sea in response to their syn-kinematic origin. They are characterized by normal fault-bounded initial synrift graben fills overlain by post-rift deposits overstepping the initial graben-like sub-basins. The full nature of the mechanical conditions for the Marmara Sea initiation are less clear. It is even partly still debated if the initiation of the Marmara Sea and the propagation of the MMF coincide in time. There are two competing hypotheses: (1) The Marmara Sea opened in extension, which weakened the lithosphere such that the North Anatolian Fault propagated along the weakened domains (e.g. Le Pichon et al., 2001; 2015) and (2) the releasing bend of the already propagated North Anatolian Fault or a dextral step-over between the MMF and the southern Fault favored local transtension resulting in the formation of the Marmara Sea as a pull-apart basin (e.g. Armijo et al., 2002; 2005). However, seismic information proves that there is a clear change in the tectonic regime with the opening of the Marmara Sea (Fig. 4: Laigle et al., 2008; Bécel et al., 2009; 2010, Bayrakci et al., 2013). The thickness between the topography–bathymetry and the base syn-kinematic sediments represents the syn-kinematic sediment fill (Fig. 3b). This thickness is on average about 2.5 km over the Marmara Sea area. Two thickness maxima indicate localized subsidence and

sediment accumulation, one aligned along the MMF where the syn-kinematic sediments are more than 5.2 km thick below the present-day Central Basin and southeastern part of the Tekirdağ Basin, and the second maximum of up to 5 km below the Çınarcık Basin limited northward by the MMF.

The depth to the Moho interface in the initial model (Fig. 2b) has been obtained by interpolating between various seismic data covering a larger area than the model area (Hergert et al., 2011, Supporting Information, Fig. S1). To constrain the Moho depth to the model area, Hergert et al. (2011) applied a Gauss filter to adjust the local variation of the Moho depth. The Moho is distinctly shallower below the Marmara Sea than below the surrounding onshore areas and shows updoming to a depth of 27 km below the basin. Along the basin margins, the Moho is about 30 km deep and descends eastwards to more than 35 km depth beneath Anatolia.

## 3.3. Geophysical data

The seismic observations considered for this study, in addition to those taken into account in the initial model, include P-wave velocity profiles from an offshore-onshore reflection-refraction survey (Bécel et al., 2009) and from a 3D seismic tomography study focused on the sediment-basement configuration of the North Marmara trough (Bayrakci et al., 2013). Both studies are based on the SEISMARMARA-Leg1 seismic survey (Hirn and Singh, 2001), and the locations of the related profiles in the model area are shown in Fig. 4a. Three-dimensional seismic tomography modelling in the North Marmara trough (Bayrakci et al., 2013) indicates that the P-wave velocities vary between 1.8 and 4.2 $km.s^{-1}$ within the syn-kinematic sediments. Bayrakci et al. (2013) derive the top of the crystalline basement as an iso-velocity surface with a P-wave velocity of 5.2 $km.s^{-1}$. In addition, relying on wide-angle reflection-refraction modelling, Bécel et al. (2009) interpreted a refractor below the base syn-kinematic sediments with a P-wave velocity close to 5.7 $km.s^{-1}$ as the top of the crystalline basement. These seismic studies suggest that the crust beneath the syn-kinematic sediments is not homogeneous as assumed in the initial model, but that there is a unit of pre-kinematic sediments beneath the syn-kinematic sediments with an average P-wave velocity of 4.7 $km.s^{-1}$ above the crystalline crust (Fig. 4). The pre-kinematic sediments encompass all deposits that have accumulated before the Marmara Sea opening. In the realm of the Marmara Sea, based on borehole observations, these deposits are separated from the syn-kinematic sediments by a diachronous unconformity that cuts units of variable age reaching from Early Cenozoic in the Upper Miocene to uppermost Cretaceous (Le Pichon et al., 2014). The pre-kinematic sediments are thinned in response to the extension/transtension related to the Marmara Sea opening that is most pronounced in the North Marmara Trough. Onshore, surface geological observations (Ergün and Özel, 1995; Genç, 1998; Turgut and Eseller, 2000; Yaltırak, 2002; Le Pichon et al., 2014) mapped Eocene–Oligocene sediments at the north-western and southern margins of the Marmara Sea that might be related to the missing units below the observed unconformity within the basin.

Furthermore, Bécel et al. (2009) interpreted a reflective horizon with a P-wave velocity of 6.7 $km.s^{-1}$ and largely parallel to the Moho topography as the top lower crystalline crust (Fig. 4b and 4c). Moreover, multichannel seismic reflection data collected in the southwestern part of the Central Basin and in the northeastern part of Marmara Island, documented a 43 km

long low-angle dipping reflector interpreted as a normal detachment fault cutting through the upper crystalline crust down to the lower crust (Fig. 4c and 4e: Laigle et al., 2008; Bécel et al., 2009). In brief, within the upper crystalline crust, the P-wave velocity varies between 5.7 km.s$^{-1}$ at the top of the crystalline basement to 6.3 km.s$^{-1}$ above the top of the lower crystalline crust. Lateral velocity variations (~ 0.3 km.s$^{-1}$) are also observed surrounding the detachment fault in the upper crystalline crust.

The first set of gravity observations considered in this study are based on EIGEN-6C4 (Förste et al., 2014). This dataset is a combined global gravity field model up to degree and order 2190 correlating satellite observations (LAGEOS, GRACE, GOCE) and surface data (DTU 2'x2' global gravity anomaly grid). We used the free-air gravity anomaly, downloaded with the resolution of ETOPO1 (1 Arc-Minute), from the International Centre for Global Earth Models–ICGEM (Barthelmes et al., 2016; Ince et al., 2019). The free-air anomaly map of the study area (Fig. 5a) displays generally low gravity values (±20 mGal) over the basin area indicating that the basin is largely isostatically compensated. An exception is a pronounced negative anomaly with values as low as -80 mGal in the northwestern area of the Marmara Sea around the MMF. Comparing the bathymetry (Fig. 1c) with the free-air gravity anomaly map, it is evident that this negative anomaly is not related to a larger basin depth as bathymetry is rather uniform along the entire axial part of the basin. Likewise, the basement of the syn-kinematic sediments (Fig. 2a) is in the same range in both sub-basins. Accordingly, the negative anomaly is not due to thickness variations of the young sediments or water depth. Apart from the onshore area next to this negative anomaly, the Marmara Sea basin is surrounded by a chain of positive free-air gravity anomalies in a range of +70 to +120 mGal that largely correlate with high topographic elevations.

The second gravity dataset used in this study is a combined satellite (TOPEX) and marine ship gravity measurements (Fig. 5b: data from Kende et al., 2017). We refer to this dataset as "Improved–TOPEX". The satellite dataset is based on a marine gravity model from CryoSat-2 and Jason-1 with the horizontal resolution of 2500 m and ~1.7 mGal of gravity accuracy over the Marmara Sea and Earth Gravitational Model 2008 over the onshore areas (EGM 2008: Pavlis et al., 2012; Sandwell et al., 2013; 2014). The shipboard gravity is from the Marsite Cruise survey in 2014 with ~1 m of horizontal resolution. Alike the gravity observations from EIGEN-6C4, this combined gravity dataset shows mostly low gravity values (±20 mGal) over the sea of Marmara and a chain of large gravity values (+70 to +120 mGal) over the onshore domain apart from the northwestern part of the model. Along the MMF, there are local negative gravity values as low as -80, -70, and -50 spatially correlating with the sub-basins of Central, Çınarcık and Tekirdağ, respectively.

The overall difference between these two datasets is a few mGals (±10 mGal), however, EIGEN-6C4 shows higher local gravity values up to 65 mGal at the southern part of the Princes Islands Segment and up to 50 mGals at the southern part of the Ganos Bend (Fig. 5c). As shown by Kende et al. (2017), the satellite gravity dataset of TOPEX has good consistency with the processed Marsite shipboard gravity data, therefore, this discrepancy is due to the different satellite gravity datasets of TOPEX and EIGEN-6C4. In summary and considering the discrepancy between the two datasets, it can be stated that apart from the local negative anomaly domains, the syn-kinematic sediments need to be isostatically balanced in the crust, given that the Moho topography is varying on a far longer wavelength below the basin.

## 4. Results

In addition to the initial structural model with a homogeneous crustal layer below the syn-kinematic sediments (Fig. 3), relying on seismic profiles (Fig. 4), we modified the structural model differentiating three crustal layers (Fig. 6). Considering the two different datasets (EIGEN-6C4 and Improved–TOPEX) and the non-uniqueness in potential field modelling, a range
of possible configurations were tested of which we present three possible best-fit models obtained from the 3D forward gravity modelling. These results are summarized in Table 1. The gravity response of these 3D structural density models and their corresponding residual gravity anomaly for each of the two gravity datasets are shown in Fig. 7 and 8, respectively.

### 4.1. Initial model

The initial model (Hergert and Heidbach, 2010; Hergert et al., 2011) resolves only the three structural units: water, syn-
kinematic sediments, and a homogeneous crust (Fig. 3). Hergert et al. (2011) considered a depth-dependent density gradient based on seismic velocities for the sediments and crust. The gradient profile varies between 1700 to 2300 kg.m$^{-3}$ within the syn-kinematic sediments, between 2500 to 2700 kg.m$^{-3}$ for the first 20 % of the crust, and from 2700 to 3000 kg.m$^{-3}$ for the lower parts of the crust. According to this profile, we derived thickness-weighted average densities of 2000 and 2800 kg.m$^{-3}$ for the syn-kinematic sediments and the crust, respectively.

The calculated gravity response of the initial model (Fig. 7a) indicates a significant misfit with respect to the observed gravity of EIGEN-6C4 (Fig. 5a). In the eastern part of the model, the misfit between observed and modelled gravity is rather small and ranges between ±20 mGal (Fig. 8a). Furthermore, within the offshore domain, along the MMF, there are two local positive residual gravity anomalies with more than +90 mGal ("A" and "B" in Fig. 8a). These positive anomalies indicate mass deficits in the model and spatially correlate with the bends along the MMF: one occurs in the southern part of the
Princes Islands Segment, between the Tuzla Bend and the Istanbul Bend, and the other one is present south of the Ganos Bend. There is also a local short-wavelength positive residual anomaly, reaching values higher than +60 mGal at the location of the Imralı Basin ("C" in Fig. 8a). In addition, a pronounced West-East oriented continuous negative residual anomaly of around -50 mGal is detected adjacent to the southern coastline.

The gravity response of the initial model shows a better fit with the observed gravity of Improved–TOPEX compared to
EIGEN-6C4 (Fig. 8b). In the onshore domain, the residual anomalies are very similar to the residual anomalies for the EIGEN-6C4 dataset. Offshore, a distinct West-East oriented continuous positive residual anomaly of around +40 mGal is noticeable along the MMF for the Improved–TOPEX dataset. In addition, two local positive residual gravity anomalies of "A" and "B" (Fig. 8a) are evident up to +60 mGal for the Improved–TOPEX dataset. The short-wavelength positive residual anomaly of "C" previously observed across the Imralı Basin (Fig. 8a) is also evident for the Improved–TOPEX dataset but
with a lower value of residual gravity up to +40 mGal.

Overall, these residuals for both gravity datasets indicate that the long-wavelength gravity field is reproduced by the initial model and that the Moho topography (Fig. 2b) is consistent with observed gravity. However, the large residual anomalies of

a few tens of km in diameter indicate the presence of crustal density heterogeneities causing gravity anomalies of smaller wavelengths, i.e. shallower depth.

## 4.2. Differentiated crust

In addition to this indication of density heterogeneities in the crust from gravity, also seismic observations (e.g. Laigle et al., 2008; Bécel et al., 2009; 2010; Bayrakci, 2009; Bayrakci et al., 2013) point to crustal heterogeneity expressed as distinct lateral and vertical variations in seismic velocity (Fig. 4). To integrate the outcomes of the seismic studies, we differentiate the crust in the next step into three units: (1) a unit of pre-kinematic sediments, (2) a unit of upper crystalline crust, and (3) a lower crystalline crustal unit.

### 4.2.1. Pre-kinematic sediments

In the initial model (Hergert and Heidbach, 2010; Hergert et al., 2011), the upper limit of the crust below the syn-kinematic sediments (their "top-basement") was mainly defined as pre-kinematic Cretaceous limestone (Ergün and Özel, 1995; Parke et al., 2002; Le Pichon et al., 2014): a surface corresponding to an increase of P-wave velocity to values larger than 4.5 km.s$^{-1}$. Furthermore, Bécel et al. (2009) interpreted a top crystalline basement as a surface where P-wave velocity increases to values above 5.7 km.s$^{-1}$ based on seismic imaging. In addition, Bayrakci et al. (2013) derived the top of the crystalline crust at an iso-velocity surface of 5.2 km.s$^{-1}$ based on a 3D P-wave tomography model beneath the North Marmara Trough. These seismic observations justify the differentiation of an additional unit of pre-kinematic sediments. Accordingly, we implement a unit the upper limit of which corresponds to the top of the pre-kinematic Cretaceous limestone (=base syn-kinematic sediments in the initial model) while its base corresponds to the top crystalline basement (Fig. 6).

The top crystalline crust topography proposed by Bécel et al. (2009) and by Bayrakci et al. (2013) is similar, and the depth difference between the surfaces presented in the two studies is mostly less than 2 km (Fig 4c). Therefore, we derive the geometry of the top crystalline basement for the gravity test applying a convergent interpolation between the seismic profiles (Fig. 4) of Bayrakci et al. (2013) and of Bécel et al. (2009).

As the newly implemented pre-kinematic sedimentary unit represents the Pre-Marmara Sea deposits, it is mostly absent in the realm of the present-day Marmara Sea (Fig. 6a). Its thickness displays maxima of up to 7.2 km along the north-western and southern margins of the present-day Marmara Sea and significantly decreases eastwards to less than 1.5 km.

Bayrakci et al. (2013) showed that the average velocity of the pre-kinematic sediments is around 4.7 km.s$^{-1}$. To convert the velocity information for this unit into density, we use an empirical equation (Eq. 1) which is a polynomial regression to the Nafe–Drake Curve valid for P-wave velocities between 1.5 to 8.5 km.s$^{-1}$ (Fig. S2 in the Supplement: Brocher, 2005 after Ludwig et al., 1970). Correspondingly, an average density of 2490 kg.m$^{-3}$ has been assigned to the pre-kinematic sediments, considering an average P-wave velocity of 4.7 km.s$^{-1}$.

$$\rho \ (g.cm^{-3}) = 1.6612V_p - 0.4721V_p^2 + 0.0671V_p^3 - 0.0043V_p^4 + 0.000106V_p^5 \qquad \text{(Eq. 1)}$$

### 4.2.2. Crystalline crust

Apart from the unit of pre-kinematic sediments, the P-wave velocity model of Bécel et al. (2009) differentiates an additional crustal interface across which P-wave velocities increase from values of around 6.2 km.s$^{-1}$ above the interface to values higher than 6.7 km.s$^{-1}$ below. They interpreted this interface as the top of the lower crystalline crust. Consequently, we applied a convergent interpolation between the seismic profiles (Fig. 4) of Bécel et al. (2009) to derive the top lower crystalline crust implemented into the next model. Eventually, we considered the thickness between the top crystalline basement and the top of the lower crystalline crust as the upper crystalline crustal unit. Its thickness distribution (Fig. 6b) shows pronounced thickness minima below the thickness maxima of the syn-kinematic sediments, where the upper crystalline crust is less than 12 km thick. In contrast, the upper crystalline crust is up to 23 km thick below the south-western margin of the present-day Marmara Sea and reaches more than 25 km in thickness along the eastern margin.

Below the upper crystalline crust, a lower crystalline crustal unit is modelled, bounded its base by the Moho discontinuity. It is characterized by an almost uniform thickness distribution (Fig. 6c) of around 10 km across the Sea of Marmara. In the north-western corner of the model area, where the Moho surface (Fig. 2b) descends, the thickness of the modelled lower crystalline crust reaches its maximum of up to 14 km. In contrast, this unit thins to less than 5 km below the south-western and north-eastern margins of the present-day Marmara Sea, where the upper crystalline crust thickens to 23 and 25 km, respectively. Offshore, adjacent to of Armutlu Peninsula, the lower crystalline crust has an increased thickness (up to 13 km) correlating with the upper crustal thinning to around 12 km.

Throughout the upper crystalline crustal unit, seismic velocities increase with depth from 5.7 km.s$^{-1}$ to 6.3 km.s$^{-1}$ (Bécel et al., 2009). Therefore, we considered 6 km.s$^{-1}$ as the average P-wave velocity of the upper crystalline crust. P-wave velocities for the lower crystalline crust show less variation, thus, 6.7 km.s$^{-1}$ has been adopted as the average P-wave velocity within the lower crystalline crust. The density for both crystalline crustal layers is calculated respecting the P-wave velocities (Eq. 1) as 2720 kg.m$^{-3}$ and 2890 kg.m$^{-3}$ for the upper and lower crystalline crust, respectively.

The gravity calculated for this refined model shows a better fit with the observed gravity datasets in comparison to the initial model (Fig. 8). Nevertheless, regarding the EIGEN-6C4 dataset, the three local large positive residual gravity anomalies observed for the initial model ("A", "B", and "C" in Fig. 8a) are still evident, indicating that the implemented subdivision of the crust alone is insufficient (Fig. 8c). The wavelength of the two other positive residual anomalies at "A" and "B" is too large to be caused by a high-density feature at the sedimentary fill level but too small to be a result of density heterogeneities in the mantle. Thus, we concluded that these misfits are most likely related to high-density bodies within the crystalline crust. The short-wavelength positive anomaly at location "C" could be interpreted as a local lack of mass within the modelled sedimentary fill of the Imralı Basin.

In contrast, considering the Improved–TOPEX dataset, implementing the pre-kinematic sediments and two crystalline crustal units instead of a uniform crustal unit successfully compensate the local positive residuals of "C" over the Imralı Basin as well as the West-East oriented continuous positive residual anomaly along the MMF (Fig. 8d). However, the residual map

still shows values of negative anomalies down to -60 mGal across the Marmara Island, in the northeast of the Kapidag Peninsula (offshore), and over the Armutlu Peninsula (D, E, and F in Fig. 8d). In addition, up to +50 mGal of positive residual anomalies are detected in the north-eastern margin of the Marmara Sea and across the Tekirdağ Basin (G and H in Fig. 8d).

### 4.2.3. Best-fit models

To overcome the remaining misfits between modelled and observed gravity, we incorporated additional crustal density heterogeneities during forward gravity modelling that we tested with respect to both gravity datasets. The gravity response of the best-fit models and their corresponding residuals are shown in Figure. 7c-e and Figure 8e-g, respectively. Over most of the model area, the residual gravity anomaly (Fig. 8e-g) shows differences between modelled and observed gravity datasets of ±20 mGal. Achieving this fit required the implementation of two dome-shaped high-density bodies of considerable dimension in the crystalline crust. Considering the differences between the two alternative gravity datasets (Fig. 5) and non-uniqueness of the gravity method, several configurations of these high-density bodies are plausible that differ in size or density. Here, we present three possible endmembers of the high-density bodies respecting both gravity datasets: one model for EIGEN-6C4 (Model-I) and two models for Improved–TOPEX (Model-II and Model-III).

4.2.3.1 Best-fit model to EIGEN-6C4 (Model-I)

In this best-fit model, high-density bodies have an average density of 3150 kg.m$^{-3}$, being thus denser than the lower crystalline crust (average density 2890 kg.m$^{-3}$), but less dense than the mantle (3300 kg.m$^{-3}$). They extend from the Moho upward, cutting through the lower crystalline crust, and reaching into the upper crystalline crust as shallow as ~5 km depth. Accordingly, the high-density bodies attain thicknesses of up to 25 km (Fig. 9 and 10).

The position of these high-density bodies spatially correlates with the domains where the MMF bends (Fig. 9 and 11). At the western margin of the Marmara Sea and below the Ganos Bend, the high-density body cuts the lower crystalline crust at a depth of around 22 km b.s.l. and continues through the upper crystalline crust. The shallower part of this body (less than 6 km b.s.l.) is located directly east of the Ganos Bend, where the MMF changes its strike direction from E-W to ENE-WSW. Likewise, the second high-density body is modelled beneath the Princes Islands segment at the eastern margin of the Marmara Sea, and the top of the body is located at a depth of around 5 km b.s.l. (Fig. 9 and 11).

By introducing the two high-density bodies into the structural model, eventually, the thickness distribution of the upper and lower crystalline crust has changed below the Çınarcık and Tekirdağ basins, where the high-density bodies largely replace the crystalline crustal units (Fig. S3 in the Supplement). Over the rest of the model area, the thickness distribution of the crystalline crustal units is similar to the one in the model explained in Sect. 4.2.2. Remarkably, the long axis of the eastern high-density body follows the strike direction of the Princes Islands segment (Fig. 9 and 11). In addition, a spatial correlation is evident between the location of the two high-density bodies with the position of the young depo-centres of the Çınarcık

and Tekirdağ basins as indicated by deepest present-day bathymetry and by thickness maxima of the syn-kinematic sediments (Fig. 1c and 3).

4.2.3.2 Best-fit models to Improved –TOPEX (Model-II and Model-III)

As shown earlier, the model with the differentiated crustal units (Fig. 6) already represents a good fit to Improved-TOPEX
(Fig. 8d). Here, we quantify the influence of the high-density bodies with an average density of 3150 kg.m$^{-3}$ on the gravity response. The forward gravity modelling output indicates that the high-density bodies need to be smaller in size for the same average density value (Model-II). The corresponding misfit between Model-II and observed gravity of Improved–TOPEX shows that the positive residuals of "G" and "H" are considerably reduced as well as the continuous negative residuals at the southern margin of the Marmara Sea (Fig. 8f). Comparing with Model-I (the best-fit model to EIGEN-6C4), these high-
density bodies can be modelled for the same location but with a smaller maximum thickness of ~16 km (Fig. 9, 10, 11).

As the second endmember solution for a best-fit model to Improved-TOPEX (Model-III), we test a configuration in which the geometry of the high-density bodies is identical to Model-I (the best-fit model to EIGEN-6C4). Therefore, the Model-III has a similar structural setting as Model-I. The results show that an average density of 2890 kg.m$^{-3}$, equivalent to the value assigned for the lower crust average density, would fit the gravity response of Model-III to Improved–TOPEX dataset best
(Fig. 8g).

In summary, all three best-fit models indicate significant lateral density variation within the crystalline crust and require the presence of two dome-shaped high-density bodies that spatially correlating with the bends of the MMF with the density ranges of ~2890 to ~3150 kg.m$^{-3}$.

**5. Interpretation and discussion of the best-fit models**

The response of the best-fit gravity models (Fig. 7c-e) and their corresponding misfit (Fig. 8e-g) confirmed that the crust below the Marmara Sea is characterized by significant density heterogeneities. In summary, these models resolve six crustal units with different densities that indicate different lithological settings within the crust (Fig. 10 and Table 1).

The uppermost and youngest layer is the present-day water column (Fig. 3a) that is largest in the present-day sub-basins of the Marmara Sea and underlain by the unit of syn-kinematic sediments of the Marmara Sea (Fig. 3b). These syn-kinematic
sediments are present mainly inside the Marmara Sea domain and their thickness distribution indicates a subsidence regime similar to the present-day one. The relationship between the individual sub-basins of the Marmara Sea and the course of the MMF are however different: The shape of the present-day Tekirdağ Basin is not evident in the thickness distribution of the syn-kinematic sediments, whereas the Central Basin along the MMF and the Çınarcık Basin are largely following their present-day counterparts. This indicates that the differentiation into the present-day Central and Çınarcık Basins postdates
the syn-kinematic phase of the Marmara Sea. The average density of 2000 kg.m$^{-3}$and the observed seismic velocities of 1800 to 4200 m.s$^{-1}$ (Bayrakci et al., 2013) indicate that this unit is mainly composed of poorly consolidated clastic deposits. There

is, however, little information on their precise ages; suggested time intervals for the deposition of this unit range from Late Miocene to Holocene with a longer deposition portion of the unit assigned to the interval between Pliocene and Holocene times (Le Pichon et al., 2014; 2015).

The third modelled unit is characterized by an average density of 2490 kg.m$^{-3}$ and by observed seismic velocities of 4200-5200 m.s$^{-1}$ (Fig. 4d and 4e: Laigle et al.,2008; Bayrakci et al., 2013) representative for sediments. At the same time, the unit is largely missing below the present-day Marmara Sea. We, therefore, interpret this unit as a Pre-Marmara Sea sedimentary unit above the top crystalline basement. The areas where the maximum thickness of more than 6 km are modelled for the pre-kinematic sediments (NW and S of the Sea of Marmara) coincide spatially with the location where Pre–Neogene rocks are present according to surface geology (Yaltırak, 2002). Other surface geological observations (Ergün and Özel, 1995; Genç, 1998; Turgut and Eseller, 2000; Le Pichon et al., 2014) also report the presence of Eocene–Oligocene sediments at the location where the maximum thickness of the pre-kinematic sediments unit is modelled.

The sedimentary units are underlain by the upper crystalline crust, which is thinned below both the Marmara Sea and the pre-kinematic sediments. This indicates that upper crustal thinning accompanied both phases of basin evolution. Both, the modelled average density and observed seismic velocities for the upper crystalline crust indicate that this unit is dominantly composed of felsic crystalline rocks. A comparison of the average density of 2720 kg.m$^{-3}$ and average P-wave velocity of 6000 m.s$^{-1}$ (Bécel et al., 2009) of the upper crystalline crust with a velocity–density pairs derived from laboratory measurements (Christensen and Mooney, 1995) indicates a composition corresponding to phyllites and/or biotite gneisses.

Below the upper crystalline crust, the lower crystalline crust follows, the top of which is largely parallel to the Moho topography. The thickness of this unit (Fig. 6c) indicates no clear spatial relationship with the formation of both generations of pre- and syn-kinematic basins. Here, the modelled average density and observed seismic velocities are indicative for an intermediate to mafic composition. Combining the physical properties of the lower crystalline crust ($\rho$ = 2890 kg.m$^{-3}$ & $V_p$= 6700 m.s$^{-1}$) and the property compilations of Christensen and Mooney (1995), the lithology of the lower crustal unit could be interpreted as diorite and/or granulite.

The sixth unit is the one with the largest differences in density-geometry configuration based on the forward gravity modelling to the two alternative gravity datasets. For this unit, we predict three alternative lateral density configurations that all entail two dome-shaped high-density bodies within the crystalline crust: two models with an average density of 3150 kg.m$^{-3}$ (Model-I and Model-II) and one model with an average density of 2890 kg.m$^{-3}$ (Model-III).

### 5.1. High-density bodies of 3150 kg.m$^{-3}$ (Model-I and Model-II)

In the best-fit gravity model with respect to EIGEN-6C4 (Model-I), the sixth unit encompasses two high-density bodies rising from the Moho in a dome-shaped manner through both crystalline crustal layers (Fig. 9a). For these bodies, a rather high density (3150 kg.m$^{-3}$) has to be assumed which indicates that they are of mafic composition. Considering the seismic velocity and density relationship (Eq. 1), a corresponding average P-wave velocity for such a high-density body with an average density would be around 7.5–7.6 km.s$^{-1}$.

In contrast, the forward gravity modelling with respect to Improved–TOPEX (Model-II) predicts the sixth unit with the same average density value of 3150 kg.m$^{-3}$ to be smaller in size (Fig. 9b). In both solutions, the locations of the high-density bodies correlate spatially with the location of two major bends of the MMF (Fig. 9 and 11) indicating that such a mafic composition in concert with their considerable thickness could result in greater strength compared to the surrounding felsic upper crust or the intermediate-mafic lower crust.

The mechanisms and timing of the emplacement of the high-density bodies are, however, difficult to determine. The modelled density indicates that the high-density bodies represent magmatic additions to the Marmara crust, potentially originating from larger depths that rose buoyantly into domains of local extension. Magnetic anomalies across the Sea of Marmara indicate positive anomalies along the MMF that may be interpreted as magnetic bodies along the fault (Ates et al., 1999; 2003; 2008). In particular, the locations of the high-density bodies beneath the Çınarcık Basin correlate spatially with the maximum positive magnetic anomaly (Ates et al. 2008) which indicates that some mafic lithology is present there below the non-magnetic sediments.

The spatial correlation between the position of the high-density bodies and the position of the eastern thickness maxima in the syn-kinematic sediments indicates that subsidence in the syn-kinematic basins at least partly took place in response to cooling of previously emplaced (magmatic) high-density bodies. This would imply that the emplacement of the high-density bodies predates the formation of the Marmara Sea sub-basins and the propagation of the MMF. To assess the possible contribution of thermal cooling to the subsidence history of the Marmara Sea, a detailed subsidence analysis with determination of the tectonic subsidence would be required.

As we do not have further evidence for a magmatic origin of the high-density bodies, other possible interpretations of these domains may be considered. For example, these high-density bodies could represent inherited structures of former deformation phases such as ophiolites along the Intra-Pontide suture that has been mapped on land, but have not yet been explored offshore (Okay and Tüysüz, 1999; Robertson and Ustaömer, 2004; Le Pichon et al., 2014; Akbauram et al., 2016). The two different emplacement mechanisms would have opposing consequences for the propagation of the North Anatolian Fault. The magmatic origin would be consistent with crustal weakening in these domains, whereas the ophiolite origin would imply the opposite. In both cases, however, a local strength anomaly in these domains would be the consequence that could be related to the bending of the fault. Whatever the origin of these bodies, their mafic composition would imply that they represent domains of higher strength in the present-day setting.

### 5.2. High-density bodies of 2890 kg.m$^{-3}$ (Model-III)

In Model-III as the alternative best-fit model for the Improved–TOPEX gravity dataset, the sixth unit has been calculated identical to the geometry of Model-I (Fig. 9a) but with the average density of 2890 kg.m$^{-3}$ as similar to the average density of the lower crust. This density value is consistent with the average density value of intermediate to mafic metamorphic rocks such as granulite (Christensen and Mooney, 1995). In this case, these two dome-shaped bodies may be interpreted as trapped

metamorphic rocks along the Intra-Pontide suture zone that spatially correlates with the North Anatolian Fault propagation (Şengör et al., 2005; Le Pichon et al., 2014; Akbauram et al., 2016).

Several studies of exhumed orogen related strike-slip faults indicate that dome-shaped metamorphic bodies of lower crust are a common phenomenon below transtensional pull-apart basins (Leloup et al., 1995; West and Hubbard, 1997; Jolivet et al.,

2001; Labrousse et al., 2004; Corsini and Rolland, 2009;). Thus the high-density bodies could represent metamorphic core complexes exhumed in response to strike-slip deformation. Such exhumation has also been proposed from numerical modelling studies across strike-slip basins such as the Sea of Marmara or the Dead Sea (Sobolev et al., 2005; Le Pourhiet et al., 2012; 2014).

### 5.3. Comparison with published 3D density model

In a previous density modelling study, Kende et al. (2017) inverted the long-wavelength gravity signals to derive the Moho topography below the Marmara region using the same Improved–TOPEX gravity dataset that we used in our study. We also consider the same bathymetry and the same seismic dataset within the Marmara Sea as Kende et al. (2017). The main difference between their density modelling and ours consists of the applied gravity methods. In our approach, we applied forward gravity modelling method while Kende et al. (2017) mainly used an inversion method to compensate the misfit

between modelled and observed gravity. The second principal difference is that Kende at al. (2017) considered the Moho depth as the primary reason for the misfit. As mentioned earlier (4.1. Initial model), the depth to the Moho in our model (Fig. 2b) has been obtained based on various seismic data covering a larger area than the Marmara region (Hergert et al., 2011, Supporting Information, Fig. S1) and was kept fixed during the forward gravity modelling. In contrast, the Moho topography in Kende et al. (2017) was obtained by gravity inversion.

We have tested the full density model of Kende et al. (2017) and the results are presented as supplementary information (Fig. S4 and S5). The misfit between the previous model (Kende et al, 2017) and the observed gravity of EIGEN-6C4 (Fig. S5 in the Supplement) generally has the same characteristics as the misfit between our differentiated crust model (two sediments units / upper crust / lower crust) and EIGEN-6C4 observed gravity (Fig. 8c). This indicates that the two positive residual anomalies of "A" and "B" (Fig. 8) are not related to the sediment thickness. Specifically, it means that the local Moho uplifts

in the model of Kende et al. (2017) would need to be much larger than 5 km to fit the calculated gravity if one considered the observed gravity datasets of EIGEN-6C4.

Comparing our results with the ones from Kende et al. (2017) we see consistent features. In particular, there is a need in both studies for a deep compensation of the sedimentary fill and Kende et al. (2017) propose to solve this with an uplift of the Moho in the domains of our lower crustal high-density bodies. In detail, assuming a laterally uniform density of the

crystalline crust, they propose ~5 km local shallowing of the Moho. In other words, Moho uplifts in their model are also high-density bodies that are 5 km thick with a density of 3330 kg.m$^{-3}$ which is comparable to ~16 km thick high-density bodies with an average density of 3150 kg.m$^{-3}$ or ~25 km thick high-density bodies with an average density of 2890 kg.m$^{-3}$ in our models.

## 5.4. Model limitations

The modelled upper and lower crystalline crustal units are consistent with seismic observations and velocity modelling (Fig. 4 and Fig. 10: Laigle et al., 2008; Bécel et al., 2009; 2010; Bayrakci et al., 2013). In contrast, seismic studies did not report the presence of large high-velocity bodies that would coincide spatially with the modelled high-density bodies. There are

only a few indications from seismic tomography (Bayrakci et al., 2013) discriminating a zone of high P-wave velocity (Vp > 6.5 km.s⁻¹) below the top crystalline basement beneath the Çınarcık Basin (Fig. 4). This high-velocity zone approximately correlates with the top of the high-density body in this area (Fig. 10). In addition, other tomography results (Yamamoto et al., 2017) indicate a zone of higher S-wave velocity and slightly higher P-wave velocity at about 20 km depth b.s.l., in the area where the western high-density body cuts the boundary between the upper and the lower crystalline crust.

While the aeromagnetic maps (Ates et al., 2003; 2008) indicate a clear positive anomaly (indicative for a mafic body at depth) beneath the Çınarcık Basin that spatially correlates with the eastern high-density bodies, there are no such indications for the western high-density body beneath the Ganos Bend. Considering the non-uniqueness of solutions in potential field modelling, other possible solutions based on different initial models should also be contemplated beneath the Ganos Bend (e.g. Kende et al., 2017; see Fig. S4 and S5 in the Supplement).

The gravity responses of the best-fit models present a good fit (±20 mGal) over most of the model area. Nevertheless, there are still some negative residual gravity anomalies across the Marmara Island, in the northeast of the Kapidag Peninsula (offshore), and over the Armutlu Peninsula ("D", "E", and "F" in Fig. 8). The short-wavelengths of these negative residual anomalies indicate that shallow low-density features remain unresolved in the model. Regarding the negative residuals anomaly at location "E", an interpretation remains difficult due to the offshore location of the anomaly. In contrast,

considering the surface geological observations might help to reveal the negative residual at the location of the Marmara Island and the Armutlu Peninsula. The thickness distribution maps (Fig. 3 and 6) show that Marmara Island is dominantly exposing rocks of the upper crystalline crust. More precisely, geological surface observations in this area (Aksoy 1995; 1996; Attanasio et al., 2008; Karacık et al., 2008; Ustaömer et al., 2009) differentiate three main rock types in outcrops: A Permian Marble unit in the North, an Eocene Granodiorite unit in the centre, and a Permian metabasite in the South of

Marmara Island. Considering the residual anomalies (Fig. 8), these three units have densities that are different from the average density assumed for the upper crystalline crust (2720 kg.m⁻³). Our result of obtaining a negative residual indicates that the subsurface extent of rocks with densities lower than the assumed average for the upper crystalline crust is larger than that of the units with higher densities. In other words, the marbles would make a larger portion of the island's subsurface than the metabasites or granodiorites.

The negative residual anomaly at Armutlu Peninsula ("F" in Fig. 8) is found where the syn-kinematic sedimentary unit is absent (Fig. 3b), whereas a thickening of the pre-kinematic sediments is modelled there (Fig. 6a). Geological maps (Genç, 1998; Yaltırak, 2002; Akbayram et al., 2016) show that this area is mainly covered by Pre-Neogene basement, Miocene acid-intermediate volcanic rocks, and some Pliocene–Holocene clastic sediments. However, the model does not account for these

locally documented occurrences of syn-kinematic sediments (Pliocene–Holocene clastics) and of Miocene volcanic rocks in this domain, which overall could explain the negative residual anomaly.

## 5.5. Implications

The gravity modelling demonstrates that considering a homogenous crystalline crust beneath the Sea of Marmara is not a valid assumption, but that rather a two-layered crystalline crust crosscut by two large local high-density (3150 kg.m$^{-3}$) bodies is plausible.

An interesting finding is the spatial correlation between the position of the high-density bodies and the two major bends of the MMF. If the high-density bodies represent high-strength domains of the Marmara Sea crust, it would cause local stress deviations influencing the fault propagation direction. The 3D view of the MMF in relation to the position of the high-density bodies illustrates how the MMF bends in these high-strength domains (Fig. 11). This would imply that the emplacement of the high-density bodies also predates the propagation of the North Anatolian Fault into the Marmara Sea. Such an interpretation would support the previously proposed hypothesis that the NAFZ reached the eastern part of the present-day Marmara Sea (Izmit) around 4 Ma before present, when the area was a domain of distributed (trans)tensional deformation, and started to propagate beneath the present-day Sea of Marmara as the MMF about 2.5 Ma ago (Le Pichon et al., 2014; 2015).

Another implication from density modelling is that the compositional and therefore also rheological heterogeneity of the Marmara crust may result in a differential response of the area to present-day far-field stresses. Accordingly, conclusions drawn from earlier studies investigating the stress-strain state in the region of the Marmara Sea with a geomechanical-numerical model (Hergert and Heidbach, 2010, 2011; Hergert et al., 2011) need to be revised.

One of the important discussions in the area of the Marmara region is on aspects that govern the dynamics of the MMF, where a 250-year lasting seismic gap 15 km south of Istanbul is observed. The western segment of the MMF is considered as a partially creeping segment (Schmittbuhl et al., 2016; Bohnhoff et al., 2017b; Yamamoto et al., 2019), whereas the eastern-central segment of the MMF is thought to be locked down to 10 km depth (Bohnhoff et al., 2013; 2017b; Ergintav et al., 2014; Sakic et al., 2016). The reasons why this seismic gap of the MMF has not ruptured over the past 250 years are debated. The felsic to intermediate crustal composition deduced from our gravity model would favour creep between the two crustal high-density bodies, whereas the two domains of the high-density bodies could represent locked segments that would require high-stress levels to fail. In case of failure, however, the energy would probably be released in a strong earthquake. These high-density bodies are interpreted as mafic and therefore represent stronger material than the surrounding felsic to intermediate crustal material of the same depth. Such rheological heterogeneities would explain the distribution of different deformation modes with creeping segments in the felsic to intermediate crustal domains and locked to critically stressed segments in the mafic domains. This hypothesis could have implications for hazard and risk assessment in this area, but need to be tested by geodynamic models considering thermo-mechanical principles.

## 6. Conclusions

In this study, 3D crustal density configurations are presented for the Sea of Marmara that integrate available seismological observations and are consistent with observed gravity. Testing successively models of increasing complexity, three best-fit models are derived that resolve six crustal units with different densities (Table 1). From our results we conclude:

(1) The present-day seafloor of the Marmara Sea has a more complex structure than during the phase of its initiation and is structured into the three main depocentres of the Tekirdağ Basin, the Central Basin, and the Çınarcık Basin.

(2) Below the present-day seafloor, the unit of syn-kinematic sediments of the Marmara Sea indicates that two main depocentres were subsiding during the early phase of basin formation. A lower sedimentary unit is interpreted as pre-kinematic sediments of the Marmara Sea. The sedimentary units are underlain by a felsic upper crystalline crust that is significantly thinned below the basin. The lowest crustal layer of regional extent is an intermediate to mafic lower crystalline crust. Both crystalline crustal layers are cut by two up-doming high-density bodies that rise from the Moho to relatively shallow depths.

(3) The emplacement of the high-density bodies within the crystalline crust could have a causal relationship with the basin-forming mechanism.

(4) The spatial correlation between the high-density bodies with two major bends of the MMF indicates that rheological contrasts in the crust may control the propagation and movement of the MMF; these high-density bodies are a possible explanation for the bends of the MMF and support the hypothesis that the MMF is geomechanically segmented.

(5) The configurations of the high-density bodies are exclusively based on 3D forward gravity modelling, a method characterized by an inherent non-uniqueness of the solutions. Only for the eastern bend, seismic and magnetic data support the presence of a deep high-density body, whereas for the western bend such indications are missing. Therefore, further geophysical observations are required to further constrain the detailed density-geometry configuration of these bodies.

(6) The high-density bodies may have an impact on the stress variability along the MMF. Thus, geomechanical models of the area should account for lateral variations in crustal density.

*Competing interests.* The authors declare that they have no conflict of interest.

*Acknowledgements.* The research leading to these results was conducted under the auspices of the *ALErT initiative (Anatolian pLateau climatE and Tectonic hazards),* an Initial Training Network (ITN) financed by the People Programme (Marie Curie Actions) of the European Union's Seventh Framework Programme FP7/2007- 2013/ under REA grant agreement no. 607996. We are indebted to Pierre Henry for his supportive comments as a referee and his contribution of the "Improved–TOPEX" gravity dataset and the structural model of Kende at al. (2017) including the high-resolution bathymetry grid. We are thankful to Mathieu Rodriguez and one anonymous referee for providing insightful reviews and constructive comments, which improved the quality of this manuscript. We also would like to acknowledge the comments of Hans-Jürgen Götze in the open discussion.

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

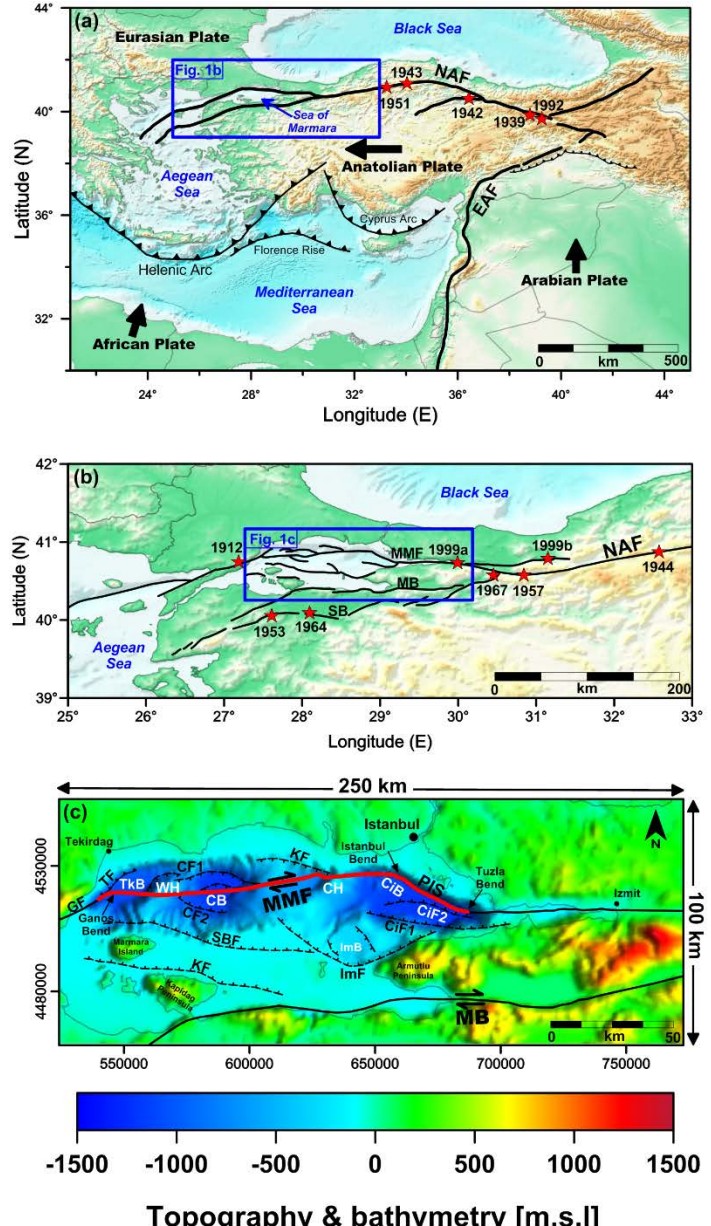

**Figure 1: Location and tectonic setting of the model area; (a) Plate tectonic map of the Anatolian plate and its relation to the Arabian, African, and Eurasian Plates; (b) The NAFZ propagation and its branches in the NW Anatolian plate; (c) The model area (WGS84 UTM Zone 35N) including the relief map of the Sea of Marmara and its surrounding onshore domain, the seismic gap since 1766 (thick red line; Bohnhoff et. al., 2017b), and the faults system identified in the investigations by Armijo et al. (2002; 2005), and Carton et al. (2007). Topography and bathymetry from ETOPO1 (Amante and Eakins, 2009) and Le Pichon et al. (2001). Abbreviations: North Anatolian Fault (NAF); Main Marmara Fault (MMF); Middle Branch of NAF (MB); Southern Branch (SB); Princes Islands Segment (PIS); Çınarcık Basin (CiB); Central Basin (CB); Tekirdağ Basin (TkB); Imralı Basin (ImB); Central High (CH); Western High (WH); Kapıdağ Fault (KF); Southern Border Fault (SBF); Imralı Fault (ImF); Çınarcık Faults (CiF 1 & 2); Kumburgaz Fault (KF); Central Basin Faults (CF 1 & 2); Tekirdağ Fault (TF); Ganos Fault (GF). Red stars show the epicentres of major earthquakes (M > 6.5) during the past century.**

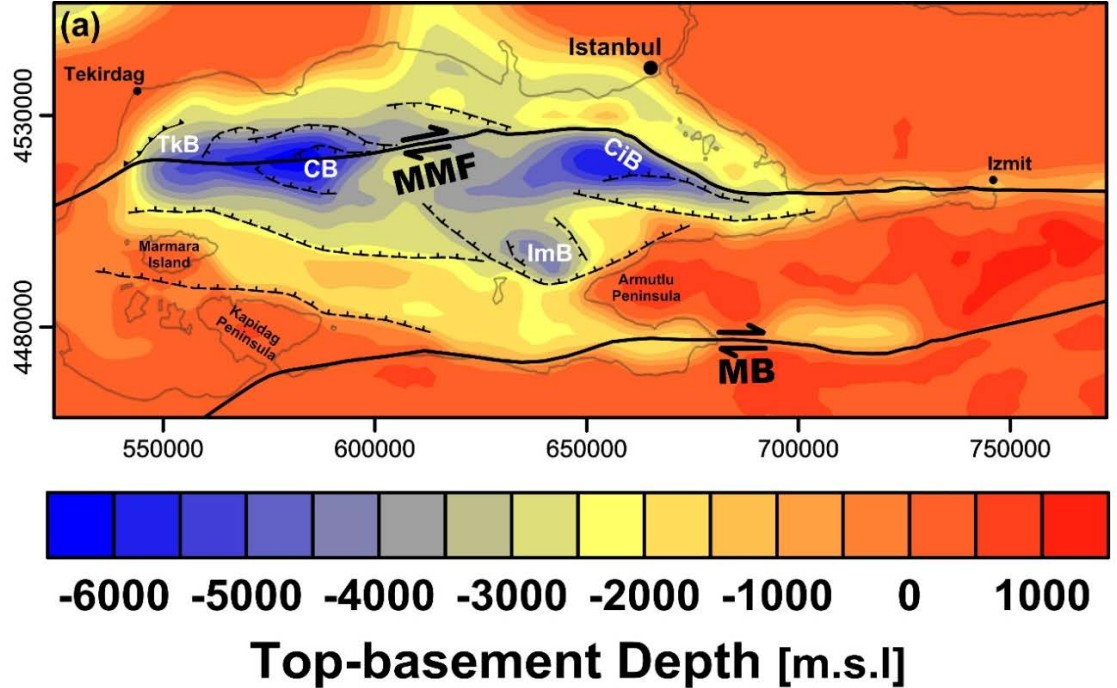

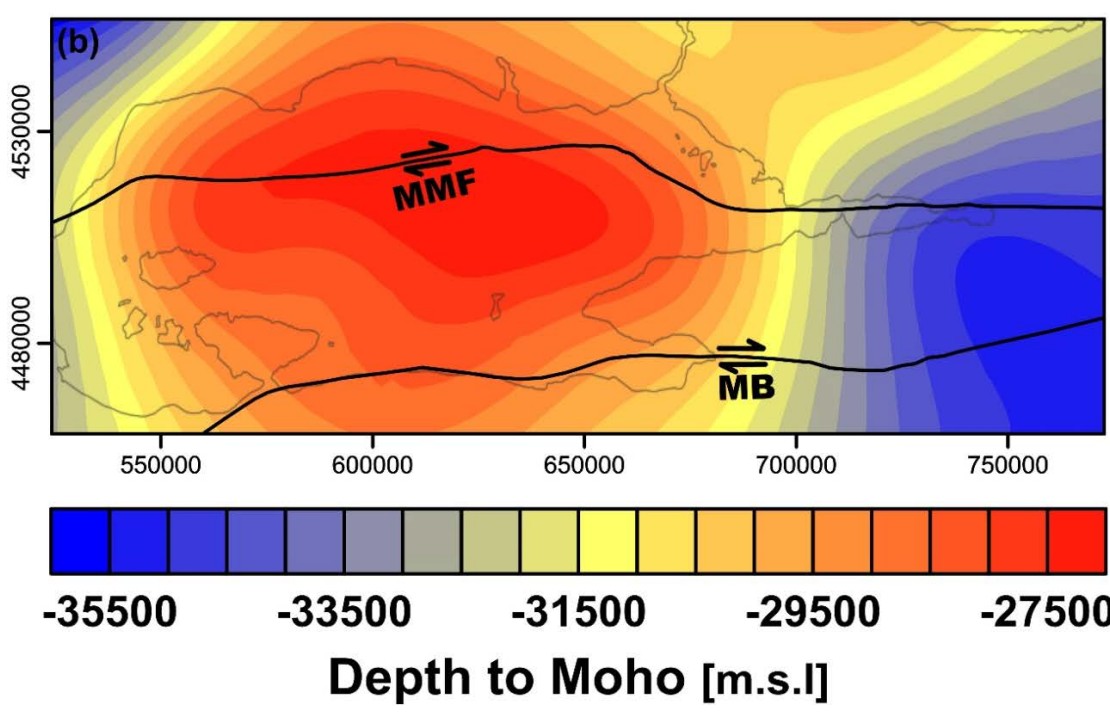

**Figure 2: Main horizons within the initial model (WGS84 UTM Zone 35N); (a) Depth to top-basement; (b) Depth to Moho. The corresponding thickness maps are illustrated in Fig. 3. Data from Hergert and Heidbach (2010). Abbreviations: Main Marmara Fault (MMF); Middle Branch of NAF (MB); Çınarcık Basin (CiB); Central Basin (CB); Tekirdağ Basin (TkB); Imralı Basin (ImB).**

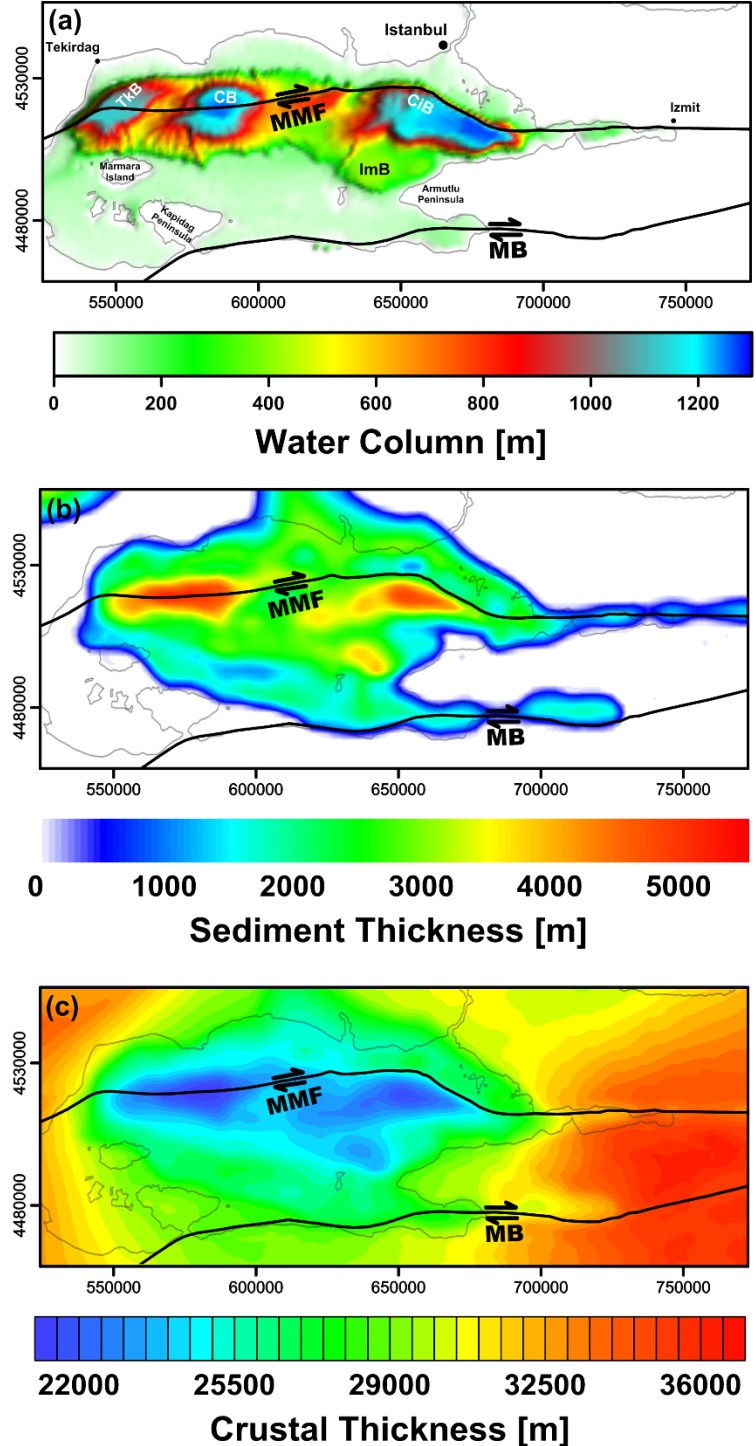

**Figure 3: Thickness distribution map of the initial structural model (WGS84 UTM Zone 35N): (a) Seawater column; (b) Syn-kinematic sediment thickness; (c) Homogeneous crustal thickness. Abbreviations: Main Marmara Fault (MMF); Middle Branch of NAF (MB); Çınarcık Basin (CiB); Central Basin (CB); Tekirdağ Basin (TkB); Imralı Basin (ImB).**

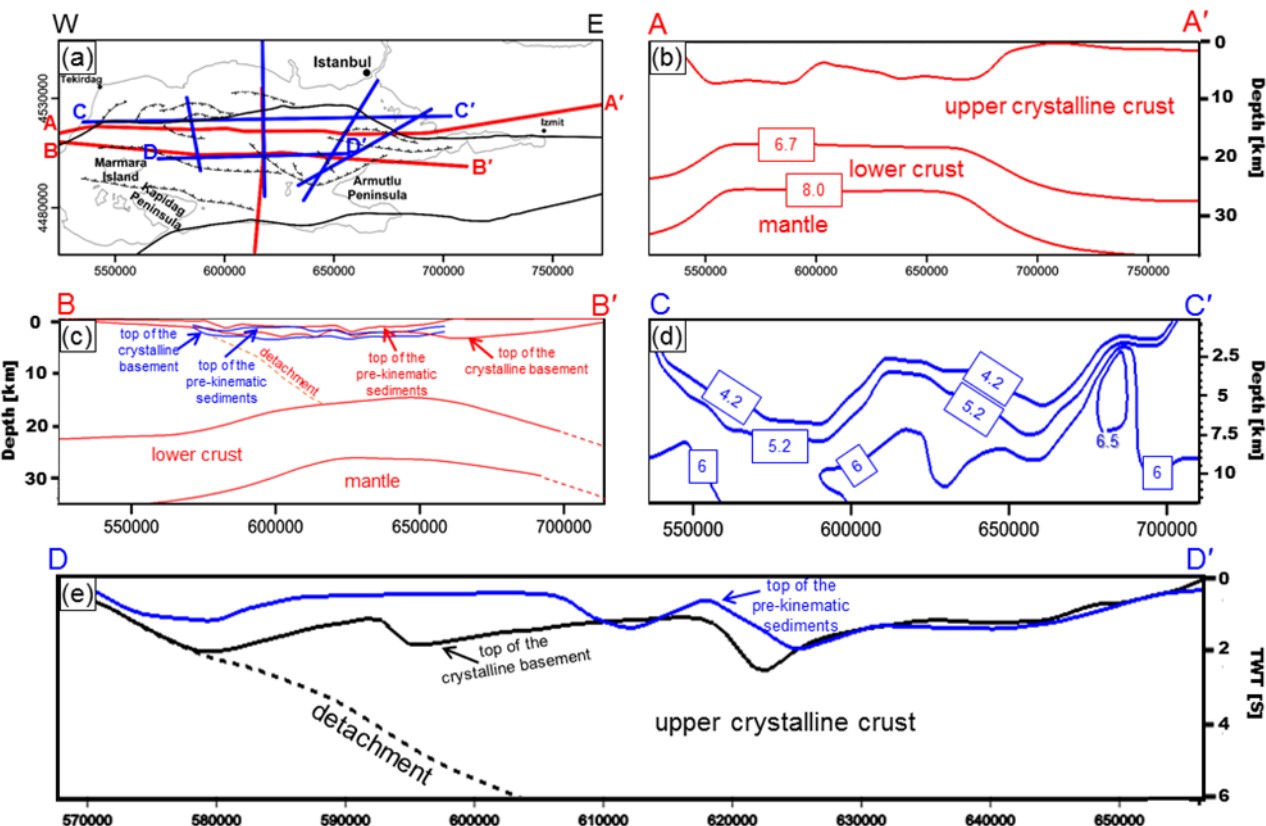

**Figure 4: Location of seismic profiles considered in this study and corresponding P-wave velocities and interpretations (modified after Laigle et al., 2008; Bécel et al., 2009; Bayrakci et al., 2013): (a) Location of the seismic profiles. Red lines are from reflection–refraction survey (Bécel et al., 2009) and blue lines are from sediment-basement tomography (Bayrakci et al., 2013); (b) Crustal structure and depth to Moho along the AA′ cross-section; (c) Crustal structure and depth to Moho along the BB′ cross-section including interpretations of the tomographic results along the DD′ profile; (d) P-wave velocity contours form the tomographic modelling along the CC′ profile; (d) Tomographic modelled isovelocity of 4.2 km.s⁻¹ (blue line) representing top of the pre-kinematic sediments in two way travel time along the DD′ profile and multichannel reflection seismic interpretation form Laigle et al. (2008) on the same profile. Numbers are modelled P-wave velocities for base syn-kinematic sediments (4.2 km.s⁻¹), base pre-kinematic-sediments (5.2 km.s⁻¹), top of the lower crust (6.7 km.s⁻¹), and the Moho discontinuity (8 km.s⁻¹).**

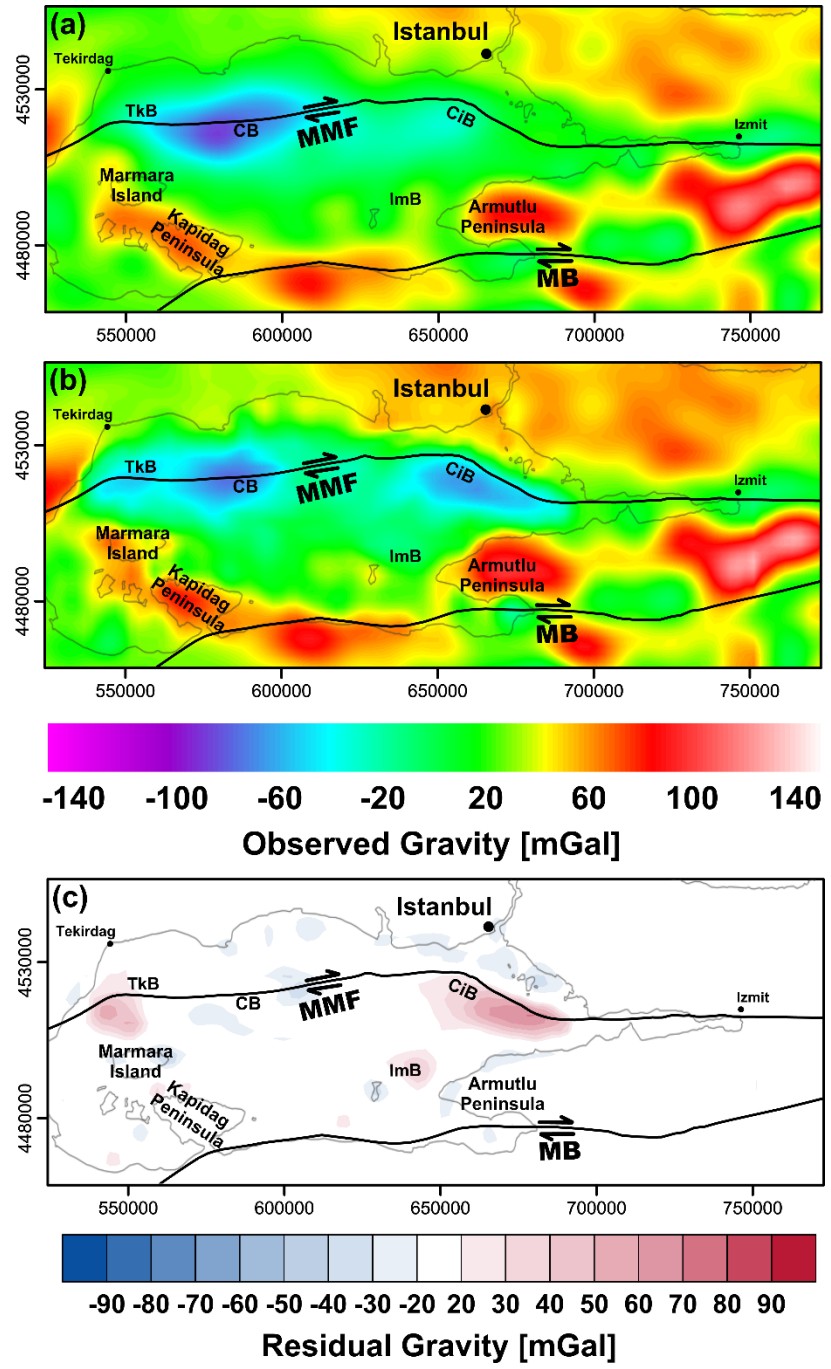

**Figure 5: Considered gravity datasets in this study (WGS84 UTM Zone 35N); (a) Observed satellite free-air anomaly (Eigen-6C4; Förste et al., 2014); (b) Free-air anomaly map of "Improved–TOPEX" from Kende et al. (2017) combining the Jason-1 and CryoSat-2 satellite data (Sandwell et al., 2014) and the Marsite cruise gravity measurements over the Sea of Marmara. Onshore gravity of this dataset is based on EGM 2008 (Pavlis et al., 2012); (c) The difference between the two gravity datasets (a - b). Abbreviations: Main Marmara Fault (MMF); Middle Branch of NAF (MB); Çınarcık Basin (CiB); Central Basin (CB); Tekirdağ Basin (TkB); Imralı Basin (ImB).**

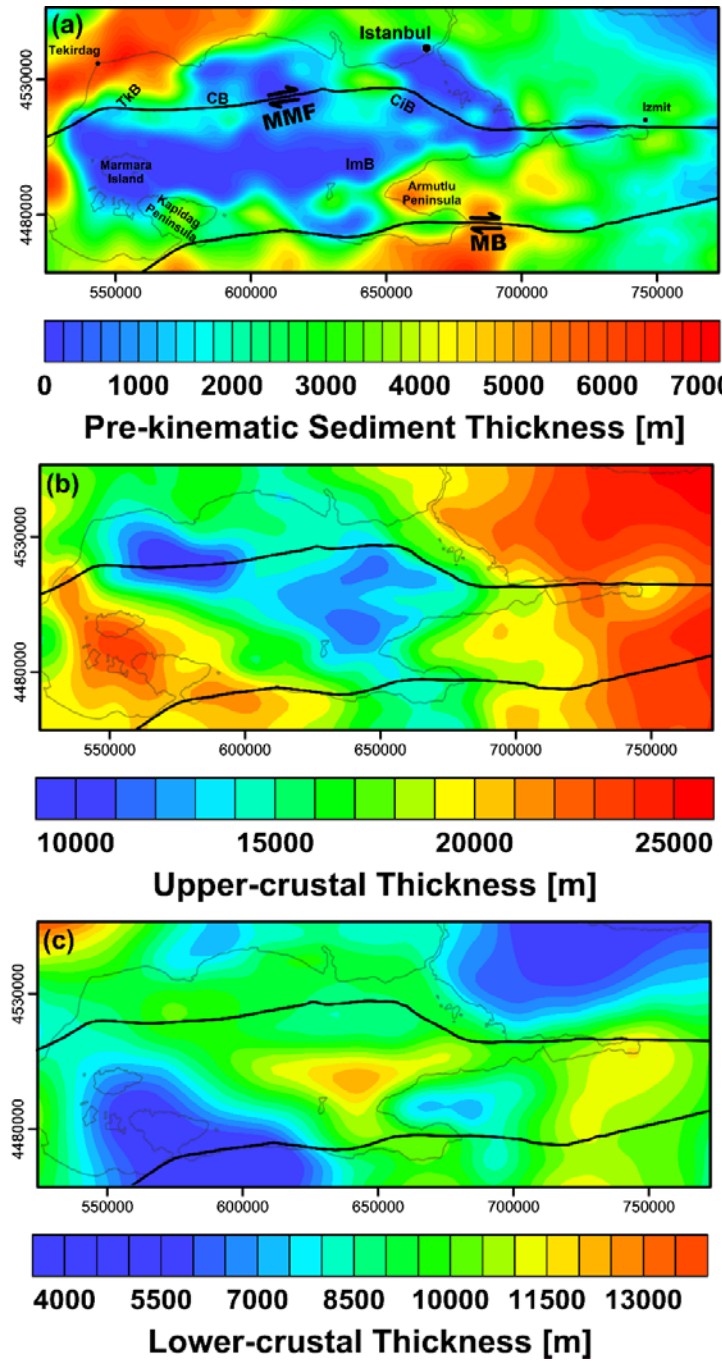

**Figure 6: Differentiated crustal structural model integrating seismic observations along the profiles in Fig. 4 (WGS84 UTM Zone 35N): (a) Pre-kinematic sediment thickness; (b) Upper crystalline crustal thickness; (c) Lower crystalline crustal thickness. Abbreviations: Main Marmara Fault (MMF); Middle Branch of NAF (MB); Çınarcık Basin (CiB); Central Basin (CB); Tekirdağ Basin (TkB); Imralı Basin (ImB).**

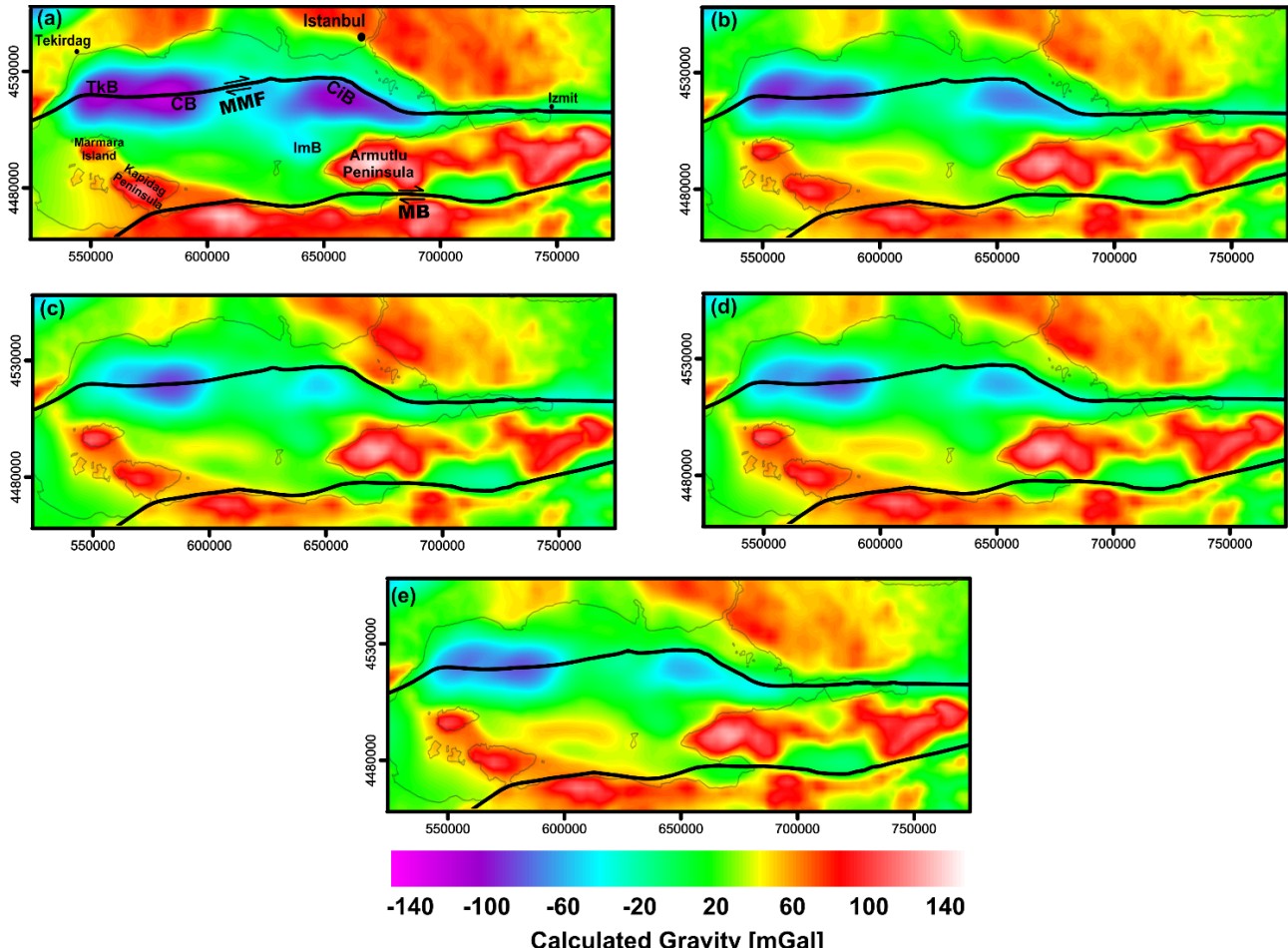

**Figure 7: Calculated gravity over the model area (WGS84 UTM Zone 35N): (a) Initial model gravity response; (b) Gravity response of a model with differentiated crust based on the seismic observations (Fig. 4); (c) Gravity response of Model-I, the best-fit model based on the forward gravity modelling on Eigen-6C4 (Förste et al., 2014); (d) Gravity response of Model-II, the best-fit model based on the forward gravity modelling on Improved–TOPEX (Kende et al., 2017); (e) Gravity response of Model-III, the alternative best-fit model based on the forward gravity modelling on Improved–TOPEX (Kende et al., 2017). The average density for the modelled high-density bodies is 3150 kg.m$^{-3}$ in Model-I and Model-II, and 2890 kg.m$^{-3}$ in Model-III. The corresponding residual gravity anomaly of each model is shown in Fig. 8.**

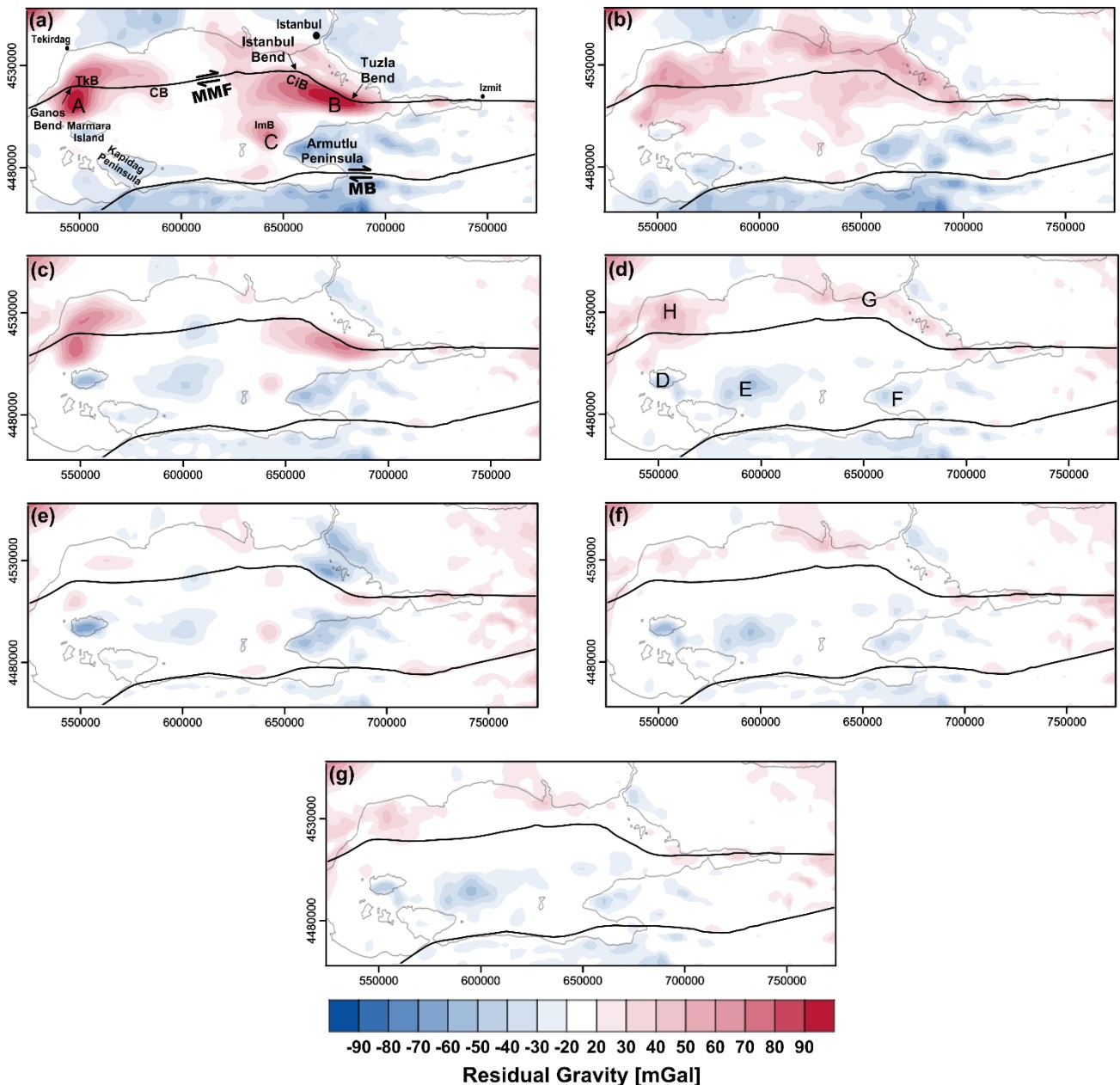

**Figure 8: Residual gravity anomaly maps show the misfit between the observed (Fig. 5) and calculated gravity (Fig. 7) of different structural model across the study area (WGS84 UTM Zone 35N): (a) Initial model to Eigen-6C4 (Förste et al., 2014); (b) Initial model to Improved–TOPEX (Kende et al., 2017); (c) Model with a differentiated crustal unit to Eigen-6C4; (d) Model with a differentiated crustal unit to Improved–TOPEX; (e) Model-I, the best-fit model based on the forward gravity modelling on Eigen-6C4; (f) Model-II, the best-fit model based on the forward gravity modelling on Improved–TOPEX; (g) Model-III, the alternative best-fit model based on the forward gravity modelling on Improved–TOPEX. The average density for the modelled high-density bodies is 3150 kg.m⁻³ in Model-I and Model-II, and 2890 kg.m⁻³ in Model-III.**

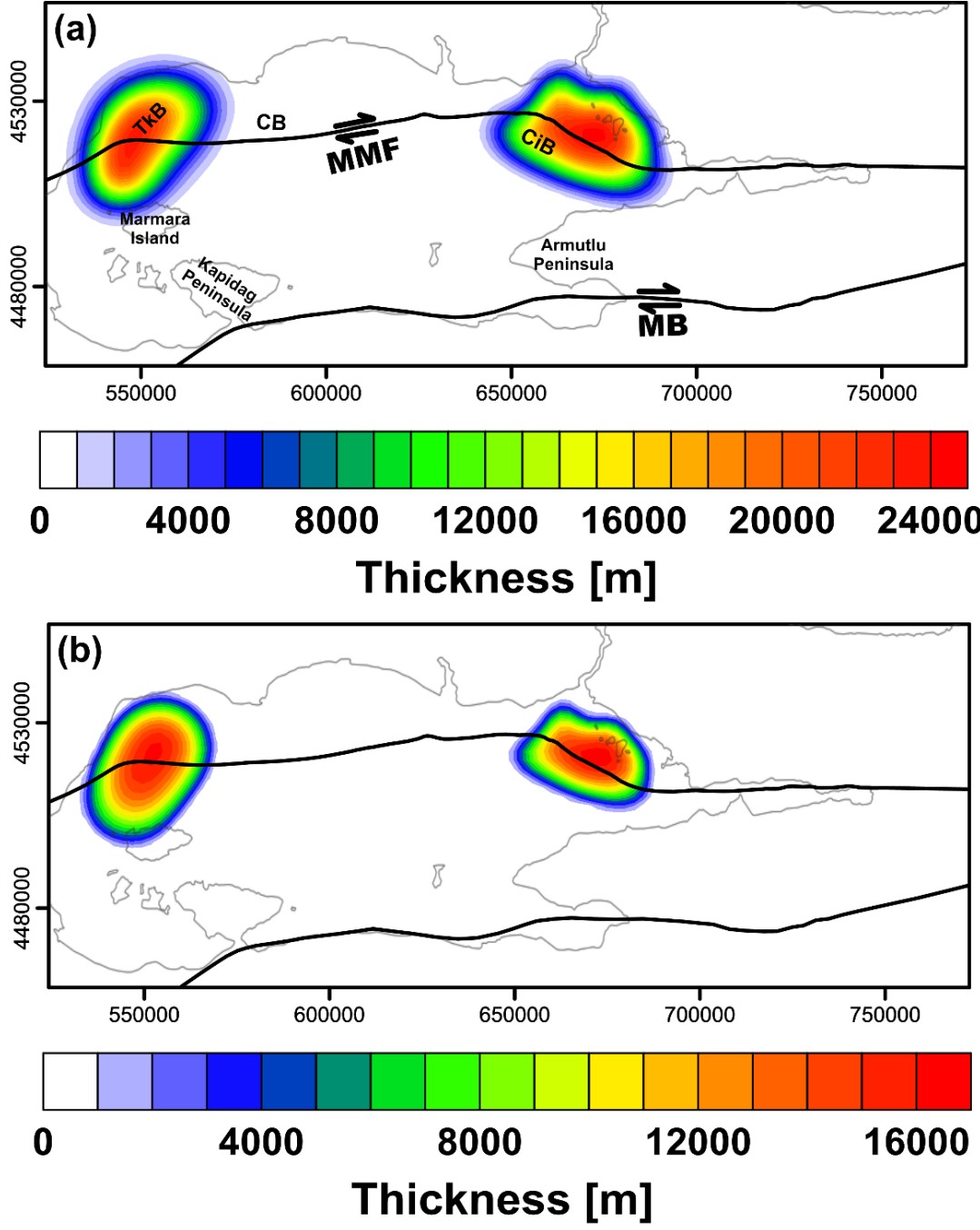

**Figure 9: Thickness of the high-density bodies achieved from the forward gravity modelling: (a) This thickness map represents the high-density bodies that present the best-fit with an average density of 3150 kg.m$^{-3}$ to EIGEN-6C4 (Model-I) and of 2890 to Improved-TOPEX (Model-III); (b) Thickness of high-density bodies with an average density of 3150 kg.m$^{-3}$ that shows the best-fit to Improved-TOPEX (Model-II).**

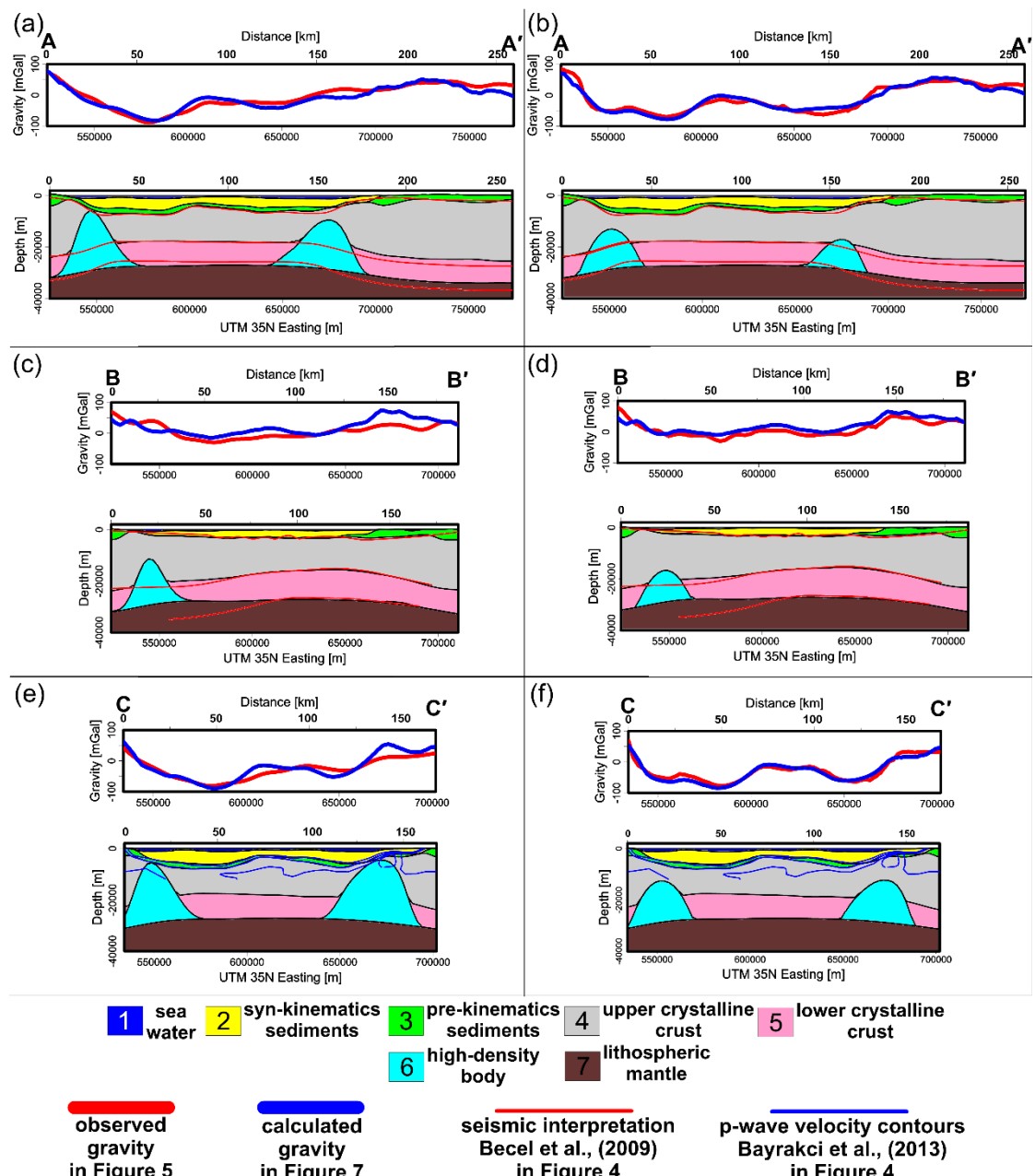

**Figure 10: Cross-sections for alternative best-fit density models to the two different gravity datasets including high-density bodies with an average density of 3150 kg.m⁻³ (Model-I and Model-II) with the observed and calculated gravity and the seismic information along the AA′, BB′, CC′ profiles in Fig. 4. Model-I shows the best-fit gravity model to EIGEN-6C4 dataset (Förste et al., 2014) and Model-II represents the best-fit gravity model to Improved–TOPEX dataset (Kende at al., 2017): (a) Model-I; (b) Model-II; (c) Model-I; (d) Model-II; (e) Model-I; (f) Model-II.**

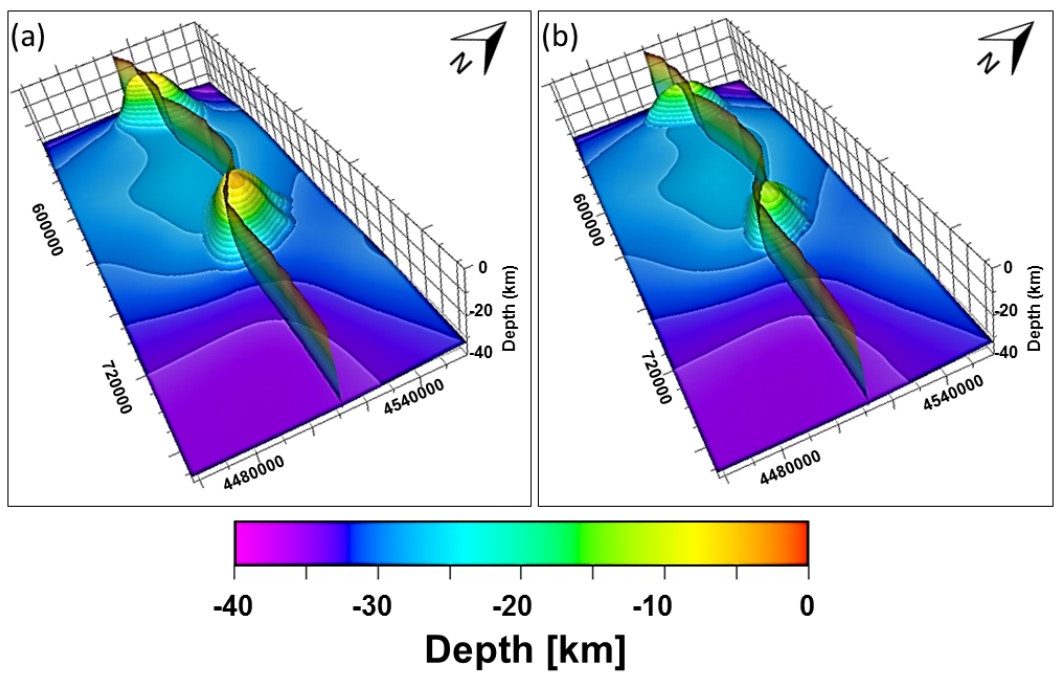

**Figure 11: 3D view of the Moho, the high-density bodies, and the MMF plane across the model area (WGS84 UTM Zone 35N). The high-density bodies location spatially correlates with the bent segments of the MMF: (a) High-density bodies according to Model-I and Model-III with an average density of 3150 and 2890 kg.m$^{-3}$, respectively; (b) High-density bodies according to Model-II with an average density of 3150 kg.m$^{-3}$. The Moho depth and the 3D fault plane from Hergert and Heidbach (2010).**

**Table 1: Structural units resolved in three alternative density models (Model-I, Model-II, and Model-III) of the Sea of Marmara with interpreted lithology and corresponding physical properties. The seismic velocity and density relationship is based on the Eq. 1 (Brocher, 2005). Note that the high-density bodies have not yet been imaged by seismic observations, and their physical properties are according to the density modelling.**

| Structural Units | Average P-wave Velocity (m.s$^{-1}$) | Average Density (kg.m$^{-3}$) | Lithological Interpretation |
|---|---|---|---|
| Seawater | − | 1025 | – |
| Syn-kinematic Sediments | 2250 (1800 to 4200)[†] | 2000 (1700 to 2300)[*] | Clastic sediments (poorly consolidated) |
| Pre-kinematic Sediments | 4700 (4200 to 5200)[†] | 2490 | Sediments (consolidated) |
| Upper crystalline crust | 6000 (5700 to 6300)[‡] | 2720 | Felsic metamorphic (biotite gneiss, phyllite)[●] |
| Lower crystalline crust | 6700[‡] | 2890 | Intermediate to Mafic (diorite, granulite)[●] |
| High-density bodies | 7550 / 6700 (?) | 3150 (Model-I & Model-II) / 2890 (Model-III) | Mafic / Intermediate to Mafic (gabbroic intrusive / diorite, granulite)[●] |
| Mantle | 8000[‡] | 3300 | – |

*†: Bayrakci et al., 2013; *: Hergert et al., 2011; ‡: Bécel et al., 2009; ●: Christensen and Mooney, 1995*