# Peer review of "3D Crustal Density Model of the Marmara Sea"

_Solid Earth, 2018_

## Referee Comment (RC1) · M. Rodriguez (Referee) · 30 Oct 2018

Review of the manuscript : 'Crustal density model of the Sea of Marmara : geophysical data integration and 3D Gravity Modelling' By Ershad Gholamrezaie et al. For Solid Earth This is a beautiful and thorough study about the crustal structure of the Marmara Sea with strong implications for the understanding of the geology of the area and the segmentation of the fault system. The study is clear, well written, with nice figures. The link with seismic and tomography studies makesyour 3D crustal model very convincing. I therefore recommend the publication in Solid Earth. However, I have a few minor comments, questions and suggestions.

[Figure]

Scientific comments: -The definition of the pre-kinematic and syn-kinematic sediments is a bit unclear and somehow difficult to relate to the complex geology of the area. What I do not understand is if this terminology refers to the timing of localization/propagation of the North Anatolian Fault, the opening of the Marmara Sea, or the onset of the Main Marmara Fault...or maybe all this stages together? I understand that the pre-kinematic sediments refers to the deposits older than Late Cretaceous, but there are also some tertiary sediments (Eocene) that were unrelated to the history of the North Anatolian Fault. Do you link these sediments with the pre or syn-kinematic history? You should dedicate a full paragraph where you explain clearly this terminology, and make a clear link with the geological episodes in this area. This terminology is sometimes confusing. -I have found the link with seismic profiles and tomography very convincing (especially the link with Becel et al. and Laigle et al.)...Maybe you should add a figure summarizing what we have learned from these studies (i.e. a few cross sections). Some readers may not be familiar with these studies and find all the related sections difficult to follow. -One of the strongest result is the identification and mapping of the high density body, with a density $\sim 3$. However I feel that the discussion about its origin is incomplete. You link these bodies to deep magmatic activity coeval with the activity of the North Anatolian Fault...but the mechanism at the origin of these high density bodies is unclear. Shear heating of the lower crust or the top of the lithospheric mantle? How can you be sure that the formation of these high density bodies is related to the activity of the North Anatolian Fault? What are the arguments? An alternative may be to consider these high density bodies reflects the intra-pontides suture zone. Parts of this suture zone has been mapped onland (see a synthesis in LePichon et al 2014), but the offshore mapping remains unclear. I wonder if what you identify may actually be some ophiolites or metamorphic rocks trapped along this suture zone. In terms of density, ophiolites are >3, some metamorphic rocks can reach the same density. For some insights about the intra-pontides suture zone, I suggest the following papers : Okay and Tüysüz, 1999; Robertson and Ustaömer, 2004. If the ophiolites/suture zone hypothesis is correct, then it means that structural inheritance strongly controls the

segmentation of the North Anatolian Fault in this area. It would also strongly emphasize some previous suggestions of Celal Sengör, who proposed that the localization of the North Anatolian Fault is strongly influenced by the intra-Pontides suture zone.

Detailed comments: -the title reads a bit long: I suggest something like '3D crustal density model of the Marmara Sea' -Geological setting : lateral escampe of Anatolia is not only the result of Arabia indentation, there is also a link with the retreat of the Hellenic trench, see Faccenna et al 2006 EPSL for an elegant synthesis -Geological setting: page3 Line 25. LePichon et al 2003 provide some observations suggesting the present-day context is pure strike slip, not transtensional (no oblique extensive stresses), except in the area of Cinarcik where the bend of the fault favors extension. . . -In the discussion, please compare better the improvements of your study with previous ones (Kende. . .etc. . .)

Comments related to the figures: -In the captions, please refer to the meaning of the abbreviations, it is sometimes boring to jump from one figure to another to find the significance. -In figure 8 : you should number the layers to ease the link with the text (for instance, when you refer to the third layer, the reader has to guess which one is it on the figure. . .) -As mentioned earlier, maybe adding some cross sections from previous works (Laigle et al 2008 especially) may help the understanding of your study for a broader audience

I hope you will find these comments helpful and constructive Best regards, Dr. Mathieu Rodriguez Ecole normale supérieure de Paris

Suggested References: Faccenna, C., Bellier, O., Martinod, J., Piromallo, C., Regard, V., 2006. Slab detachment beneath eastern Anatolia : a possible cause for the formation of the North Anatolian Fault. Earth and Planetary Science Letters 242, 85-97. Okay, A.I., Tüysüz, O., 1999. Tethyan sutures of northern Turkey. Geological Society of London, Special publications, 156, 475-515. Robertson, A.H.F., Ustaömer, T., 2004. Tectonic evolution of the intra Pontide suture zone in the Armutlu Peninsula, NW

[Figure]

Turkey. Tectonophysics 381, 175-209

---

## Referee Comment (RC2) · Anonymous Referee #2 · 11 Dec 2018

The paper addresses the question of the deep crustal structure of the submerged section of the North-Anatolian Fault within the Sea of Marmara, which may have important implications to better assess the earthquake hazard in the highly populated (> 15 Millions inhabitants) Istanbul area. A new crustal-scale 3D density model integrating geological and seismological data is presented, based on additional 3D-gravity modelling. The major result is that the crust appear to be Âńcrosscut by two large, dome-shaped mafic high-density bodies (average density of 3050 kg.m-3 ) of considerable thickness above a rather uniform lithospheric mantle (3300 kg.m-3 )Âż. It is to be noted here that the location of these two bodies coincides with the location of two major escarpments : below the Tekirdag and the Cinarçik escarpments, respectively (Figure 9c). As

a conclusion, the authors then suggest that these high-density bodies Âń control the rheological behaviour along the NAFZ, and consequently, influences fault segmentation and propagation dynamics Âż.

The paper is well presented and well written. However, there are major concerns regarding the dataset, both for gravimetry and for bathymetry from the offshore domain in Sea of Marmara.

1) For gravity, the authors use the EIGEN-6C4 dataset (Förste et al., 2014), which is a combined global gravity field model up to degree and order 2190 correlating satellite observations (LAGEOS, GRACE, GOCE) and surface data (DTU 2'x2' global gravity anomaly grid). At the scale and wavelengths concerned by the present study : 1) the DTU 2'x2' global gravity anomaly grid, based on satellite altimetry, is predominant and 2) the density contrast is at the sea-bottom interface is of critical importance.

It is highly regrettable that no discussion is presented to compare the free-air gravity anomaly from ship-board gravimeters and the satellite derived gravity data used in the present paper for the offshore domain. In Figure 2 of Kende et al (note missing reference : J. Geophys. Res. Solid Earth, 122, 1381–1401, doi:10.1002/ 2015JB012735), the differences between the two datasets are shown along a 130 km long profile, oriented along the strike of the main fault, following the deeper parts of the Sea of Marmara. This profile represents the most favourable configuration for using gravity from radar altimetry. Still, there are major differences.

The N-S profiles (B-B' and C-C') shown in Figures 7 and 8 of the present submission represent the worst configuration for satellite altimetry-derived gravity, as they cross sharp escarpments bordering the Tekirdag and Cinarçik basins, which are expected to produce important effects on the gravity signature. A comparison between satellitre gravity and ship-board gravity must be presented and the effects related to the use of altimetry-derived gravimetry must be discussed.

2) For topography-bathymetry (shown figure 1c), the authors use a dataset exported

from 1 Arc-Minute Global Relief Model (Amante and Eakins, 2009), which integrates the 30 arc-second grid obtained from NASA's Shuttle Radar Topography Mission (SRTM) and a bathymetry dataset from the MediMap Group, 2008.

Bathymetric grids from the Medimap group have a 1 km grid-node spacing. Compared to high-resolution grids based on shipboard, multibeam echsounders (e.g. [Le Pichon et al., 2001]), such grids are expected to smooth considerably the bathymetry, when sharp escarpments are present, particularly at the Western Tekirdag and the Northern Cinarcik escarpments. A smoothen bathymetry at escarpments may induce unwanted effects in gravity modelling, by introducing artificially the need of compensating high density bodies at depth.

The concerns listed above on both the gravimetry and the bathymetry datasets, cast serious doubts on the reality of the two high density bodies found by the authors. Besides these two major issues, a geological discussion on the implications of the results is cruelly missing (gravity model solutions are not unique ; geological criteria represent the best guides for discussing non-unique solutions).

In conclusion, for the above reasons, I do not recommend publication of the submitted paper in Solid Earth Discussions.

A substantial effort is needed : 1) for testing the relevance of the gravity model they use in the case of the Sea of Marmara (particularly due to the presence of sharp escarpments) 2) for testing the relevance of the bathymetric grid 3) for presenting an in-depth, geological discussion for discriminating the different (non-unique) results.

Please also note the supplement to this comment:
https://www.solid-earth-discuss.net/se-2018-113/se-2018-113-RC2-supplement.pdf

---

## Author Comment (AC1) · 20 Dec 2018

Dear members of the Editorial Board,

We thank reviewer2 for his/her comments and would like to directly respond to his/her critical remarks. We agree that there is a mistake concerning the correct referencing of the Kende et al. paper and apologize for this.

The correct reference of course is:

Kende, J., Henry, P., Bayrakci, G., Özeren, M.S. and Grall, C.: Moho depth and crustal thinning in the Marmara Sea region from gravity data inversion, Journal of Geophysical

[Figure]

Research: Solid Earth, 122(2), 1381–1401, doi: 10.1002/2015JB012735, 2017.

We also thank reviewer2 for pointing us to the discrepancy between different gravity data sets. We however strongly disagree with the reasons reviewer2 gives for rejecting the paper. In the following, we address the different concerns of reviewer2.

(1) The first major concern of reviewer2 is that we have used the wrong gravity data set. In particular, the reviewer mentions a higher resolved data set introduced in the Kende et al. (2017) paper. Here we have to state that we have used the publicly available data set EIGEN-6C4 (Förste et al., 2014) because it covers the onshore and offshore parts of the study area. The higher resolved dataset the reviewer recommended to use instead and presented in Kende et al. (2017) is not publicly available. We have managed to access the mentioned satellite data shown in their Figure 2 (Sandwell et al., 2014; http://topex.ucsd.edu/WWW_html/mar_grav.html), but not the shipboard data for the offshore parts. Anyway, the "improved" data set of Kende et al. (2017) differs only by max a few mGals in the offshore domain from the satellite data (Sandwell et al., 2014), but has the same general characteristics. So the problem raised by the reviewer relates to the difference in two different versions of satellite gravity data published in the same year by two different well-known and experienced teams.

As the scope of our paper is not to quality check satellite gravity data, but to make use of it together with other observations, we decided for the EIGEN-6C4 free-air gravity anomaly data set.

Nevertheless, we thank the reviewer for pointing out this discrepancy between the different gravity data sets and we carefully have carried out a sensitivity analysis to test which difference we obtain between our model adjusted for the EIGEN-6C4 and a model adjusted to the data set of Sandwell et al. (2014). The results of this sensitivity analysis are enclosed to this response as a supplement file (Figure. 1). Accordingly, the two different satellite gravity data sets are almost identical over the onshore domains where the absolute difference is less than 10 mGal. Offshore, a notable difference of 30 to 50 mGal is evident along the bending segments of the Main Marmara Fault, the regions below which we have modelled the high-density bodies in the lower crust. It is clear that the difference between the two data sets is smaller than the gravity response of the high-density bodies. Thus we can confirm that the high-density bodies are still required, though the alternative data sets (Sandwell et al., 2014 and the one including the shipboard gravity of Kende et al., 2017) would require the high-density bodies to be slightly smaller in size/or density (non-uniqueness of gravity). The quantitative comparison enclosed shows that the top of high-density bodies would be 5 km deeper for a model consistent with the data set of Sandwell et al. (2014) than for our best-fit model consistent with the data set of Förste et al. (2014). The fit with the data set of Sandwell et al. (2014) could be even further improved by considering a density of 2990 Kg/m3 for the high-density bodies. We suggest to include this comparison in the paper in the revised version and thus document the related uncertainties. In addition, we could propose which additional (seismic) data could help to discriminate between these endmember configurations derived mainly from gravity.

Apart from the two different data sets, there are consistent findings in our study and the study of Kende et al. (2017). In particular, the latter also shows the need for deep compensation of the sedimentary fill, however, the authors propose to solve the problem with an uplift of the Moho in the domains of our lower crustal high-density bodies. In detail, they propose local shallowing of the Moho – and therewith also high-density bodies that are 5 km thick with a density of 3330 kg/m3 (compared to up to 18 km of density 3050 kg/m3 in our model) –, assuming a laterally uniform density of the crystalline crust. This is confirming our results rather than discarding them.

Seismological data used for model construction (Becel et al.,2009; Hergert et al., 2011) indicate that no such pronounced Moho uplift is present in the domains of our high-density bodies, a point also admitted by Kende et al. (2017). They critically review this misfit with their model and mention uncertainties in the seismic data as possible reasons for the misfit. However, if these uncertainties in the seismological constraints

are small, the derived Moho uplift may not be there and the crystalline crust may not be as uniform as suggested by Kende et al.'s gravity modelling.

Moreover, the limited available seismological observations (Becel et al., 2009; Karabulut et al., 2013; Bayrakci et al., 2013; Yamamoto et al., 2017) indicate that seismic velocities vary within the crystalline crust, in particular, an increase in seismic velocities (Bayrakci et al., 2013) is found for the uppermost part of the high-density bodies modelled in our study. Yamamoto et al. (2017) present results from a tomography study indicating a zone of higher seismic velocities in the areas where the modeled western high-density body cuts the interface between upper and lower crystalline crust.

Finally, the locations of the lower crustal high-density bodies also correlate spatially with a positive magnetic anomaly (Ates et al. 2003) which indicates that some mafic lithology is present below the non-magnetic sediments. Thus, assuming a uniform density and a +/- constant thickness of the upper and lower crystalline crust separated by an interface running parallel to the Moho is hard to justify.

(2) The other main point of the reviewer is that we do not consider the right bathymetry, in particular, the one presented in Kende et al. (2017) or in more detail in Le Pichon et al. (2001). Here we would like to clarify that this is a question of the horizontal resolution of our model. As we analyze a lithosphere-scale model, its horizontal resolution cannot resolve small-scale details. Nevertheless, we consider the main characteristics of the bathymetry, in particular, the location of maximum seafloor depth, only the steepness of the present-day basin margins is not resolved to the same level of accuracy. More specifically the reviewer claims that not properly considering the steep slopes resolved in the high-resolution bathymetry may question our general results concerning the deep crustal structure. This is simply impossible because such differences would result in a few mGals maximum difference with very local and short wavelength characteristics in the gravity response, but would not question the presence or absence of deep bodies causing a response of at least several tens of mGals and tens of km in wavelength. This can be demonstrated easily with a comparison of the grav-

ity response of respective models. We therefore clearly disagree with the reviewer's judgement concerning this point.

In summary, considering the higher resolved bathymetry and the higher resolved gravity data may help in defining sharper boundaries of the high-density bodies but would not question their presence and therefore our main results.

(3) The reviewer also asks for more discussion of the geological implications of our results. We indeed have been rather reluctant here as we wanted to avoid going too far with the interpretation, but could easily provide a few lines of thoughts here. We can add discussion on the consequences of the different interpretations for the deep structure of the Marmara Sea against the background of previously proposed concepts. This would touch hypotheses for the deformation mechanism that created the Marmara Sea and for the present day distribution of strength in the crust.

We hope these arguments will convince the editors that the work is worth publishing and will contribute to increasing our understanding of continental transform faults.

Best regards,

Ershad Gholamrezaie

On behalf of all co-authors

Please also note the supplement to this comment:
https://www.solid-earth-discuss.net/se-2018-113/se-2018-113-AC1-supplement.pdf

**Supplement:**

[Figure]

*Figure 1. Sensitivity test for two different gravity data sets over the Sea of Marmara: (a) satellite free-air anomaly from EIGEN-6C4 (Förste et al., 2014); (b) satellite free-air anomaly from CryoSat-2 and Jason-1 (Sandwell et al., 2014); (c) difference between the two data sets (a-b); (d) residual gravity between EIGEN-6C4 and best-fit model of Gholamrezaie et al. (2018) with the top of the high-density bodies at around 5 km depth b.s.l (Fig. 7c in submitted paper); (e) residual gravity between satellite data of Sandwell et al. (2014) and best-fit model of Gholamrezaie et al. (2018) with the top of the high-density bodies at around 4 km depth b.s.l; (f) residual gravity between satellite data of Sandwell et al. (2014) and best-fit model of Gholamrezaie et al. (2018) with the top of the high-density bodies at around 9 km depth b.s.l. MMF stands for Main Marmara Fault.*

---

## Editor Comment (EC1) · Mandea (Editor) · 4 Jan 2019

Dear authors,

The discussion for this manuscript has been re-opened so that the referees can submit their additional comments (received by email during the winter holidays). They have been informed accordingly.

Best wishes, Mioara MANDEA (handling topical editor)
* * *

---

## Referee Comment (RC3) · Henry (Referee) · 7 Jan 2019

This manuscript presents an interesting new hypothesis explaining gravity anomalies in the Sea of Marmara area: the presence of high density bodies within the crust along the North Anatolian fault zone. However, the manuscript does not yet provide a fully convincing demonstration that the presence of these bodies is required by the available data.

Owing to the non-uniqueness of gravity inversion solutions, and to the limitations of the currently available constraints from seismology, the gravity modeling alone cannot prove the existence of the high density bodies. Data may also be fit (at least at

wavelengths of more than about 30 km) considering relatively small variations of Moho depth that remain compatible with constraints from seismology. The presence of high density bodies, is, however, a sound hypothesis, which can be further supported by considering the geological and geophysical contexts.

Geological knowledge on the Sea of Marmara area is already integrated in the discussion, but two important points are missing: (1) Ates et al. (1999, 2003, 2008) found magnetic anomalies in the Sea of Marmara area, which they related to the presence of magnetic bodies along the North Anatolian Fault zone. The largest one coincides with the eastern dense body infered in this study. (2) The North Anatolian fault zone follows more or less an ophiolitic suture, and this could explain at least in part the presence of dense and/or magnetic bodies along its track. Heterogeneities in the crust may thus not be a consequence of magmatic intrusions during a rifting event, but be a consequence of the convergent, and then transcurrent, tectonics during the Paleogene. This is already appearent in some of the cited references (e.g. Sengor et al., 2005) and more recent references also exist (e.g. Akbauram et al., 2016).

My conclusion would be that the gravity anomaly in the Eastern Sea of Marmara is at least in part caused by a mafic/ultramafic sliver in the crust, but it is still unclear to me whether a large high density body is present beneath Tekirdag Basin. I fully agree with the authors that these bodies could be a possible factor controling strain localization within the North Anatolian shear zone and that they predate the Pio-Quaternary transtensional tectonics, but I am not convinced they were emplaced as magnatic intrusions within the continental crust.

Regarfing the discussion with Reviewer #2, I would like to confirm that the Sandwell/TOPEX gravity model has good consistency with the marine data that were collected during Marsitecruise (both used in Kende et al., 2015), and that the Eigen-6C4 anomaly map used here seems less consistent with these marine data. I would like to encourage the authors to go on with their suggestion to compare models fitting Topex and Eigen-6C4 gravity anomalies. I would be happy to provide the gravity data

used in Kende et al. to the authors (hence, do not request to stay anonymous). Ideally, a magnetic model could be added.

References :

Akbayram, K., Şengör, A. M. C., & Özcan, E. (2016). The evolution of the Intra-Pontide suture : Implications of the discovery of late Cretaceous – early Tertiary mélanges. In R. Sorkhabi (Ed.), Tectonic Evolution, Collision, and Seismicity of Southwest Asia: In Honor of Manuel Berberian's Forty-Five Years of Research Contributions: Geological Society of America Special Paper 525 (Vol. 525). https://doi.org/10.1130/2016.2525(18)

Ates, A., Kayiran, T., & Sincer, I. (2003). Structural interpretation of the Marmara region, NW Turkey, from aeromagnetic, seismic and gravity data. Tectonophysics, 367, 41–99. https://doi.org/10.1016/S0040-1951(03)00044-1

Ates, A., Kearey, P., & Tufan, S. (1999). New gravity and magnetic anomaly maps of Turkey. Geophysical Journal International, 136(2), 499–502. https://doi.org/10.1046/j.1365-246X.1999.00732.x

Ates, A., Bilim, F., Buyuksarac, A., & Bektas, Ö. (2008). A tectonic interpretation of the Marmara Sea, NW Turkey from geophysical data. Earth, Planets and Space, 60(3), 169–177. https://doi.org/10.1186/BF03352780

---

## Short Comment (SC1) · 8 Jan 2019

First of all I would like to thank the authors for their stimulating manuscript about 3D constrained gravity modelling in/around the Marmara Sea, as well as for the comments of the two reviewers. It is important to note first of all that the 3D modelling was done in a group with a high international standing in geophysical modelling of static as well as dynamic and kinematic processes. The same applies to their competence in geological interpretations of complex areas.

Their paper describes the results, the data processing, constraining information used and the 3D modelling process in this endangered area very competently and precisely.

[Figure]

The software tools used are based on proven and recognized numerical methods and procedures. The 3D lithospheric density model is constrained by seismic tomography and geological findings. Gravity data are based on the Förste et al. (2014) gravity model EIGEN6C4 and the 1 arc. min. global relief model. These are data sets which were extremely helpful in the modelling and interpretation of static and dynamic Earth problems - as in this case.

Against this background and a careful study of Gholamrezaie et al's manuscript, which is up for discussion, the conclusion of Reviewer 2 seems a bit overdone to me. I even find it quite unfair, as the reviewers' criticism is not really based on numerical findings made with the data sets under discussion, but on "vague visual impressions". This becomes clear in Gholamrezaie's reply, where they pointed out that differences between the two data sets in question (EIGEN 6C4 and Kende et al. 2017 data) leave no general doubt as to the validity of the published interpretation. There are two other observations concerning the "Kende et al. 2017" gravity data set and the EIGEN 6C4: (1) the reviewer 2 mentioned a data set which is not fully available to the public and (2) After the study of the Kende et al. publication several aspects remain unanswered regarding the achieved accuracy/uncertainty in data processing (topographic correction) and 3D modelling.

In my opinion the manuscript should be published soon, since it contains concepts which contain valuable hints for future work in further works (3D modelling). A rejection, as demanded by Reviewer 2, is not really justified - and my evaluation corresponds rather to the vote of Reviewer 1.

Finally, I would like to conclude a very personal remark based on my long experience as an author and reviewer of international journal articles. As a reviewer, I have always been careful not to "rewrite" the manuscript available for the review: the decisive factors for the quality of a manuscript are the handling of the data/information, the actuality of the data processing and numerical interpretation as well as its evaluation. In this sense the manuscript in question can be accepted - in this case with references to the

otherwise still available data sets and their quality.

Hans-Jürgen Götze, Institut für Geowissenschaften, CAU Kiel, Geophysik
* * *

---

## Referee Comment (RC4) · Anonymous Referee #2 · 14 Jan 2019

In my initial review, I recommended rejection because of concerns regarding the datasets (gravity and bathymetry) used by the authors in the submarine domain of the Sea of Marmara. This issue

Ershad Gholamrezaie and his co-authors claim that they have used the only publicly available data set EIGEN-6C4 (Förste et al., 2014) that covers the onshore and offshore parts of the study area, as the shipboard gravity data is supposedly not available. Pierre Henry (Reviewer #3) mentioned that he will be happy to make this data available.

Hence, my new recommendation is that the authors should be given a chance to revise

their paper, using all the marine geophysical data (shipboard gravity and bathymetry) that were collected over the last years in the offshore domain.

In his reply, Ershad Gholamrezaie mentions out a number of points that he takes for granted, as listed below:

1. "Anyway, the "improved" data set of Kende et al. (2017) differs only by max a few mGals in the offshore domain from the satellite data (Sandwell et al., 2014), but has the same general characteristics. So the problem raised by the reviewer relates to the difference in two different versions of satellite gravity data published in the same year by two different well-known and experienced teams. "

I am not convinced, Ershad Gholamrezaie must prove this allegation. The profile shown by Kende et al (Profile XX', shown in Figure 2) strikes in the E-W direction, along the central part of the Sea of Marmara, where the altimetry-derived gravity grid is likely to be optimum. The problem arises along N-S profiles, particularly in the western part of the Tekirdag Basin, where only a few satellite data points are available in the N-S direction.

2. "As the scope of our paper is not to quality check satellite gravity data, but to make use of it together with other observations, we decided for the EIGEN-6C4 free-air gravity anomaly data set".

I do not agree, by no means, with this approach. The very first, most important thing the authors should do, is to make sure that the dataset they use is appropriate. EIGEN-6C4 is a global dataset, it is undoubtly the best one for global studies. For local studies, however, there may be significant discrepancies leading to unwanted artefacts. The "local" vs "global" grids is a well known problem in geophysics and the authors should carefully go into it.

3. "We have carried out a sensitivity analysis to test which difference we obtain between our model adjusted for the EIGEN-6C4 and a model adjusted to the data set

of Sandwell et al. (2014). The results of this sensitivity analysis are enclosed to this response as a supplement file (Figure. 1). Accordingly, the two different satellite gravity data sets are almost identical over the onshore domains where the absolute difference is less than 10 mGal. Offshore, a notable difference of 30 to 50 mGal is evident along the bending segments of the Main Marmara Fault, the regions below which we have modelled the high-density bodies in the lower crust."

This is well expected, of course, because the two models are based on space observations. Comparison should be made between satellite altimetry-based and shipboard gravity-based data. Here again, the authors must prove thoroughly their allegations. The authors suggest to include this comparison in the paper in the revised version and thus document the related uncertainties, including seismic data: this is very good idea, that should be encouraged.

4. "The reviewer claims that not properly considering the steep slopes resolved in the high-resolution bathymetry may question our general results concerning the deep crustal structure. This is simply impossible [. . . and] can be demonstrated easily with a comparison of the gravity response of respective models."

The authors should be invited to demonstrate their claims, by comparing the gravity response of the different models using different datasets, including shipboard datasets.

In conclusion : the use of global vs local grids for local studies is an issue of critical importance. The authors must demonstrate that they can successfully address this issue in the specific case of the Sea of Marmara.

---

## Editor Comment (EC2) · Mandea (Editor) · 18 Jan 2019

I suggest the authors to consider the important reviews and to use all the marine geophysical data (shipboard gravity and bathymetry) that were collected over the last years in the offshore domain.

---

## Author Comment (AC2) · 11 Mar 2019

Dear members of the Editorial Board,

First of all, we would like to thank Mathieu Rodriguez for reading our manuscript and providing us with the valuable suggestions and the detailed comments.

Please find enclosed our detailed answers to the reviewer's comments. We hope the reviewer find his concerns satisfactorily addressed and that the revised manuscript can be accepted for publication.

Yours sincerely

[Figure]

On behalf of all co-authors,

Ershad Gholamrezaie

Please also note the supplement to this comment:
https://www.solid-earth-discuss.net/se-2018-113/se-2018-113-AC2-supplement.pdf

**Supplement:**

**Comments to M. Rodriguez, Referee #1, [rodriguez@geologie.ens.fr](mailto:rodriguez@geologie.ens.fr)**

**Review of the manuscript: 'Crustal density model of the Sea of Marmara: geophysical data integration and 3D Gravity Modelling' By Ershad Gholamrezaie et al. For Solid Earth. This is a beautiful and thorough study about the crustal structure of the Marmara Sea with strong implications for the understanding of the geology of the area and the segmentation of the fault system. The study is clear, well written, with nice figures. The link with seismic and tomography studies makes your 3D crustal model very convincing. I therefore recommend the publication in Solid Earth. However, I have a few minor comments, questions and suggestions.**

We thank the reviewer for the encouraging review.

**Scientific comments:**

**The definition of the pre-kinematic and syn-kinematic sediments is a bit unclear and somehow difficult to relate to the complex geology of the area. What I do not understand is if this terminology refers to the timing of localization/propagation of the North Anatolian Fault, the opening of the Marmara Sea, or the onset of the Main Marmara Fault. . .or maybe all this stages together? I understand that the prekinematic sediments refers to the deposits older than Late Cretaceous, but there are also some tertiary sediments (Eocene) that were unrelated to the history of the North Anatolian Fault. Do you link these sediments with the pre or syn-kinematic history? You should dedicate a full paragraph where you explain clearly this terminology, and make a clear link with the geological episodes in this area. This terminology is sometimes confusing.**

We thank the reviewer for pointing this out. We have clarified the formulation to make clear what we mean with syn-kinematic/pre-kinematic sediments (see page 6 lines 8-29 and page 7 lines 15-22 to in revised MS).

With syn-/pre-kinematic we mean with respect to the opening of the Marmara Sea. How far this is related to activity of the NAFZ needs to be discussed. The thickness variation of the youngest sediment unit indicates a clear spatial relationship with respect to two points: (1) with the present-day sub-basins of the Marmara Sea as imaged by bathymetry and (2) with respect to the trace of the MMF in that the latter partly coincides with the margin of the sub-basins. On the other hand, the thickness of the pre-kinematic sedimentary unit displays pronounced minima in the domain of the present-day Marmara Sea, which indicates that this unit has been disrupted during the formation of the Marmara Sea.

**I have found the link with seismic profiles and tomography very convincing (especially the link with Becel et al. and Laigle et al.): Maybe you should add a figure summarizing what we have learned from these studies (i.e. a few cross sections). Some readers may not be familiar with these studies and find all the related sections difficult to follow.**

We agree with the reviewer and have added an example for a seismically derived structural cross-section (new Fig. 4), that we also compare to cross-sections in the same position through the three presented end-member models (new Fig. 10).

**One of the strongest result is the identification and mapping of the high density body, with a density ~3. However I feel that the discussion about its origin is incomplete. You link these bodies to deep magmatic activity coeval with the activity of the North Anatolian Fault but the mechanism at the origin of these high density bodies is unclear. Shear heating of the lower crust or the top of the lithospheric mantle? How can you be sure that the formation of these high density bodies is related to the activity of the North Anatolian Fault? What are the arguments? An alternative may be to consider these high density bodies reflects the intra-pontides suture zone. Parts of this suture zone has been mapped onland (see a synthesis in LePichon et al 2014), but the offshore mapping remains unclear. I wonder if what you identify may actually be some ophiolites or metamorphic rocks trapped along this suture zone. In terms of density, ophiolites are >3, some metamorphic rocks can reach the same density. For some insights about the intra-pontides suture zone, I suggest the following papers: Okay and Tüysüz, 1999; Robertson and Ustaömer, 2004. If the ophiolites/suture zone hypothesis is correct, then it means that structural inheritance strongly controls**

the SED segmentation of the North Anatolian Fault in this area. It would also strongly emphasize some previous suggestions of Celal Sengör, who proposed that the localization of the North Anatolian Fault is strongly influenced by the intra-Pontides suture zone.

Indeed, the discussion was rather brief in the first version of the MS and we agree that there are more concepts to include in the discussion on the origin and nature of these bodies. We thank the reviewer for pointing us to the respective literature, that we have studied. We now provide a more extended discussion on this point, also considering hypotheses put forward in previous work (see page 13, Sec 5. Interpretation and discussion of the best-fit models). We, however, prefer to stay careful in this discussion, as our results are not suited to discriminate between several possible interpretations. We therefore discuss the implications of the different possible interpretations concerning the origin of the high-density bodies but refrain from favouring one. Regarding the intra-pontides suture zone and its relation to the high-density bodies we added the following paragraphs:

Page 15 line 16: "As we do not have further evidence for a magmatic origin of the high density bodies, other possible interpretations of these domains may be considered. For example, these high density bodies could represent inherited structures of former deformation phases such as ophiolites along the intra-Pontide suture that has been mapped on land, but have not yet been explored offshore (Okay and Tüysüz, 1999; Robertson and Ustaömer, 2004; Le Pichon et al., 2014; Akbauram et al., 2016). The two different emplacement mechanisms would have opposing consequences for the propagation of the North Anatolian Fault. The magmatic origin would be consistent with crustal weakening in these domains, whereas the ophiolite origin would imply the opposite. In both cases, however, a local strength anomaly in these domains would be the consequence that could be related to the bending of the fault. Whatever the origin of these bodies, their mafic composition would imply that they represent domains of higher strength in the present-day setting."

Page 15 line 26: "In Model-III as the alternative best-fit model for the Improved–TOPEX gravity dataset, the sixth unit has been calculated identical to the geometry of Model-I (Fig. 9a) but with the average density of 2890 kg.m$^{-3}$ as similar to average density of the lower crust. This density value is consistent with the average density value of intermediate to mafic metamorphic rocks such as granulite (Christensen and Mooney, 1995). In this case, these two dome-shaped bodies may be interpreted as trapped metamorphic rocks along the Intra-Pontide suture zone that spatially correlates with the MMF propagation (Şengör et al., 2005; 30 Le Pichon et al., 2014; Akbauram et al., 2016)."

**Detailed comments:**

**the title reads a bit long: I suggest something like '3D crustal density model of the Marmara Sea'**

We agree and have changed the title accordingly.

**Geological setting: lateral escape of Anatolia is not only the result of Arabia indentation, there is also a link with the retreat of the Hellenic trench, see Faccenna et al 2006 EPSL for an elegant synthesis**

Thanks for pointing this out, we have complemented in the text accordingly:

page 2 line 25: "In the large-scale plate-tectonic framework of Asia Minor, the NAFZ accommodates the westward escape of the Anatolian plate in response to the northward motion and indentation of the Arabian plate into Eurasia and westward enlarging of the deep slab detachment beneath the Bitlis–Hellenic subduction zone (Fig. 1a: McKenzie, 1972; Şengör et al., 2005; Faccenna et al., 2006; Jolivet et al., 2013)…"

**Geological setting: page3 Line 25. LePichon et al 2003 provide some observations suggesting the present-day context is pure strike slip, not transtensional (no oblique extensive stresses), except in the area of Cinarcik where the bend of the fault favors extension:**

Scanning the debate on this issue we found that there is contradictory interpretation of the few true stress observations. Hergert and Heidbach (2011) provide plausible arguments for lateral variations in stress regime. We therefore would like to report the full spectrum of the discussion and decided to keep this statement, though

complemented by the respective reference. As suggested by the reviewer, the following paragraph was added the revised manuscript:

Page 3 line 24: "In contrast, based on GPS velocity data and surface geological observations, there are also arguments that the kinematics of the MMF correspond to a pure right-lateral strike-slip with the exception of the Çınarcık Basin area that the bend of the Princes Islands segment causes a transtensional setting (e.g. Le Pichon et al., 2003; 2015).''

**In the discussion, please compare better the improvements of your study with previous ones (Kende. . .etc...)**

This has extensively been done, see also answers to reviewers2 and 3.

**Comments related to the figures:**

**In the captions, please refer to the meaning of the abbreviations, it is sometimes boring to jump from one figure to another to find the significance.**

Done.

**In figure 8: you should number the layers to ease the link with the text (for instance, when you refer to the third layer, the reader has to guess which one is it on the figure...)**

Done as new Fig. 10.

**As mentioned earlier, maybe adding some cross sections from previous works (Laigle et al 2008 especially) may help the understanding of your study for a broader audience**

Done, new Fig 4.

**I hope you will find these comments helpful and constructive Best regards, Dr. Mathieu**

Rodriguez Ecole normale supérieure de Paris

Indeed, we found these comments helpful, thanks again.

**Suggested References:**

Faccenna, C., Bellier, O., Martinod, J., Piromallo, C., Regard, V., 2006. Slab detachment beneath eastern Anatolia: a possible cause for the formation of the North Anatolian Fault. Earth and Planetary Science Letters 242, 85-97.

Okay, A.I., Tüysüz, O., 1999. Tethyan sutures of northern Turkey. Geological Society of London, Special publications, 156, 475-515. Robertson, A.H.F., Ustaömer, T., 2004. Tectonic evolution of the intra Pontide suture zone in the Armutlu Peninsula, NW Turkey. Tectonophysics 381, 175-209.

Integrated.

---

## Author Comment (AC3) · 11 Mar 2019

Dear members of the Editorial Board,

First of all, we would like to thank reviewer #2 for providing us with the detailed comments and for revising his/her recommendation towards "major revisions" instead of "reject".

In our revised version we attempt to address all concerns raised by the reviewer and provide new results considering the full amount of accessible observations. Before we explain how we have addressed the specific concerns of the reviewer in detail

(enclosed) we would like to give a short summary of the additional work that went into the revised version of this manuscript:

We have revised the structural model by implementing the higher resolved bathymetry provided by referee Pierre Henry.

We tested the sensitivity of our results by calculating a series of "best-fit models" with respect to both gravity data sets and present a detailed discussion of these results. This quantitatively illustrates how robust the results are and in which range uncertainties are involved.

We have included the additional hypotheses concerning the origin of the modelled high-density bodies in the discussion considering the references suggested by all three reviewers.

Please find enclosed our detailed answers to the reviewer's comments. We hope the reviewer find his/her concerns satisfactorily addressed and that the revised manuscript can be accepted for publication.

Yours sincerely

On behalf of all co-authors,

Ershad Gholamrezaie

Please also note the supplement to this comment:
https://www.solid-earth-discuss.net/se-2018-113/se-2018-113-AC3-supplement.pdf

**Supplement:**

**Comments to RC2: 'Major concerns regarding the gravimetric and bathymetric datasets used in the study cast doubt on the results', Anonymous Referee #2.**

We thank reviewer #2 for helpful comments and have earlier directly responded in the open discussion. For completeness we list the main points to the reviewer's comments again here.

The paper addresses the question of the deep crustal structure of the submerged section of the North-Anatolian Fault within the Sea of Marmara, which may have important implications to better assess the earthquake hazard in the highly populated (> 15 Millions inhabitants) Istanbul area. A new crustal-scale 3D density model integrating geological and seismological data is presented, based on additional 3D-gravity modelling. The major result is that the crust appear to be crosscut by two large, dome-shaped mafic high-density bodies (average density of 3050 kg.m-3) of considerable thickness above a rather uniform lithospheric mantle (3300 kg.m-3 ). It is to be noted here that the location of these two bodies coincides with the location of two major escarpments: below the Tekirdag and the Cinarçik escarpments, respectively (Figure 9c). As a conclusion, the authors then suggest that these high-density bodies control the rheological behaviour along the NAFZ, and consequently, influences fault segmentation and propagation dynamics. The paper is well presented and well written. However, there are major concerns regarding the dataset, both for gravimetry and for bathymetry from the offshore domain in Sea of Marmara.

1) For gravity, the authors use the EIGEN-6C4 dataset (Förste et al., 2014), which is a combined global gravity field model up to degree and order 2190 correlating satellite observations (LAGEOS, GRACE, GOCE) and surface data (DTU 2'x2' global gravity anomaly grid). At the scale and wavelengths concerned by the present study: 1) the DTU 2'x2' global gravity anomaly grid, based on satellite altimetry, is predominant and 2) the density contrast is at the sea-bottom interface is of critical importance. It is highly regrettable that no discussion is presented to compare the free-air gravity anomaly from ship-board gravimeters and the satellite derived gravity data used in the present paper for the offshore domain. In Figure 2 of Kende et al (note missing reference: J. Geophys. Res. Solid Earth, 122, 1381–1401, doi:10.1002/ 2015JB012735), the differences between the two datasets are shown along a 130 km long profile, oriented along the strike of the main fault, following the deeper parts of the Sea of Marmara. This profile represents the most favourable configuration for using gravity from radar altimetry. Still, there are major differences. The N-S profiles (B-B' and C-C') shown in Figures 7 and 8 of the present submission represent the worst configuration for satellite altimetry-derived gravity, as they cross sharp escarpments bordering the Tekirdag and Cinarçik basins, which are expected to produce important effects on the gravity signature. A comparison between satellite gravity and ship-board gravity must be presented and the effects related to the use of altimetry-derived gravimetry must be discussed.

(1) The first major concern of reviewer #2 is that we have used the wrong gravity data set. In particular, the reviewer mentions the higher resolved data set introduced in the Kende et al. 2017 paper. We would like to repeat that it was not the scope of this paper to quality check published and downloadable gravity data, as these went through a review process before publication. We wanted to explore, what additional understanding can be gained if such data is integrated with previous models, other geophysical data and forward 3D gravity modelling. We agree that there is a mistake concerning the correct referencing of the Kende et al., 2017 paper and apologize for this. The reference was corrected and substantial discussion has been added to MS with respect to both the gravity data set as well as the bathymetry presented by Kende et al. (2017).

For the revised version of this paper we have tested the sensitivity of our results with respect to both data sets: Förste et al. (2014) and Kende et al. (2017). We had used the publicly available data set EIGEN-6C4 (Förste et al., 2014) in the initial submission because it covers the onshore and offshore parts of the study area. The higher resolved dataset the reviewer recommended to use instead and presented in Kende et al. (2017) was not publicly available for the initial submission. Fortunately, thanks to support from reviewer #3 P. Henry, the Kende at al. (20179 datasets was made available to us and we could extend our analysis beyond our initial reply to reviewer #2 in the open discussion.

We explored the gravity response of different model configurations with respect to both data sets and present 3 "best fit" endmember models in the revised manuscript that illustrate the sensitivity of the results. In addition, we supply further details in the Supplementary Information.

Nevertheless, we thank the reviewer for pointing out this discrepancy between the different gravity data sets and we carefully have checked which differences we obtain between our model adjusted for the EIGEN-6C4, a model adjusted to the dataset of Sandwell et al. (2014) or a model adjusted for the data set of Kende etal. (2017). We can confirm that the high-density bodies are still required, though fitting the different gravity datasets would require the high-density bodies to be slightly smaller in size or density (non-uniqueness of gravity). We have included a detailed comparison in the paper in the revised version and thus document the related uncertainties. Nevertheless, there are consistent findings in our study and the study of Kende et al. (2017). In particular, the latter also show the need for deep compensation of the sedimentary fill, however, the authors propose to achieve this implementing as uplift of the Moho in the domains of our lower crustal high density bodies. In detail, they propose local shallowing of the Moho – and therewith also high-density bodies that are 5 km thick with a density of $3330 kg.m^{-3}$, compared to +15 km of density $3000 kg.m^{-3}$ in our initial model, assuming a laterally uniform density of the crystalline crust. This is supporting our results rather than discarding them. We have added a quantitative comparison in the new manuscript in this respect (page 12: Sec 4.2.3. Best-fit models, and page 16: Sec 5.3. Comparison with published 3D density model).

Seismological data used for model construction (e.g. Becel et al., 2009) indicate that no such pronounced Moho uplift is present in the domains of our high-density bodies, a point also admitted by Kende et al. (2017). They critically review this misfit with their model and mention uncertainties in the seismic data as possible reasons for the misfit. However, if these uncertainties in the seismological constraints are small, the derived Moho uplift may not be there and the crystalline crust may not be as uniform as suggested by Kende et al.'s gravity modelling results. Moreover, the limited available seismological observations (Becel et al., 2009; Karabulut et al., 2013; Bayrakci et al., 2013) indicate that seismic velocities vary within the crystalline crust.  In particular, an increase in seismic velocities is found in the regions where the uppermost part of the high-density bodies modelled in our study are located (New Fig.4 and Fig. 10)

Finally, the locations of the lower crustal high-density bodies also correlate spatially with a positive magnetic anomaly (Ates et al., 1999;2003; 2008), also suggested to consult by reviewer #3. This indicates that some mafic lithology is present below the non-magnetic sediments. Thus, assuming a uniform density and a +/- constant thickness of the upper and lower crystalline crust separated by an interface running parallel to the Moho is difficult to justify.

In summary all evaluated gravity data sets require the presence of local bodies of higher than average crustal density in the deeper crust. If these are large and characterized by a smaller density contrast to the surrounding crystalline crust or smaller and of higher density remains unclear. Here, additional deep seismic data would help to reduce non-uniqueness.

2) For topography-bathymetry (shown figure 1c), the authors use a dataset exported from 1 Arc-Minute Global Relief Model (Amante and Eakins, 2009), which integrates the 30 arc-second grid obtained from NASA's Shuttle Radar Topography Mission (SRTM) and a bathymetry dataset from the MediMap Group, 2008. Bathymetric grids from the Medimap group have a 1 km grid-node spacing. Compared to high-resolution grids based on shipboard, multibeam echsounders (e.g. [Le Pichon et al., 2001]), such grids are expected to smooth considerably the bathymetry, when sharp escarpments are present, particularly at the Western Tekirdag and the Northern Cinarcik escarpments. A smoothen bathymetry at escarpments may induce unwanted effects in gravity modelling, by introducing artificially the need of compensating high density bodies at depth. The concerns listed above on both the gravimetry and the bathymetry datasets, cast serious doubts on the reality of the two high density bodies found by the authors. Besides these two major issues, a geological discussion on the implications of the results is cruelly missing (gravity model solutions are not unique; geological criteria represent the best guides for discussing non-unique solutions). In conclusion, for the above reasons, I do not recommend publication of the submitted paper in Solid Earth Discussions. A substantial effort is needed: 1) for testing the relevance of the gravity model they use in the case of the Sea of Marmara (particularly due to the presence of sharp escarpments) 2) for testing the relevance of

the bathymetric grid 3) for presenting an in-depth, geological discussion for discriminating the different (non-unique) results.

(2) This other main point of the reviewer is that we do not consider the right bathymetry, in particular the one presented in Kende et al. (20179 or in more detail in Le Pichon et al. (2001). Though we are sure that this would not be of primary importance, given the horizontal resolution of our lithosphere-scale model we have implemented this bathymetry into the revised models. Again we would like to acknowledge the generous supply of this data set by reviewer #3, P. Henry. The differences with respect to the initial model related to this modification were indeed in the range of a few mGals and thus do not question the presence or absence of deep bodies causing a response of at least several tens of mGals and tens of km in wavelength. Accordingly, considering the higher resolved bathymetry and the higher resolved gravity data has helped in defining sharper boundaries of the high density bodies but their presence was still required.

The reviewer also asks for more discussion of the geological implications of our results, which was also suggested by the other reviewers. We have therefore added discussion on the consequences of the different interpretations for the deep structure of the Marmara Sea against the background of previously proposed concepts (page 13-16). This indeed has sharpened the respective parts of the MS with respect to hypotheses for the deformation mechanism that created the Marmara Sea and for the present day distribution of strength in the crust.

---

## Author Comment (AC4) · 11 Mar 2019

Dear members of the Editorial Board,

We would like to acknowledge the generous help of referee Pierre Henry (reviewer #3) who not only made helpful comments but also made data available to us. He gave us access to the improved gravity data set TOPEX published by Kende et al. (2017) and to the higher resolved bathymetry of the Marmara Sea.

We have revised the structural model by implementing the higher resolved bathymetry and tested the sensitivity of our results by calculating a series of "best-fit models" with

respect to both gravity data sets and present a detailed discussion of these results.

Please find enclosed our detailed answers to the reviewer's comments. We hope the reviewer find his concerns satisfactorily addressed and that the revised manuscript can be accepted for publication.

Yours sincerely

On behalf of all co-authors,

Ershad Gholamrezaie

Please also note the supplement to this comment:
https://www.solid-earth-discuss.net/se-2018-113/se-2018-113-AC4-supplement.pdf

―――――――――――――――――

[Figure]

**Supplement:**

**Comments to P. Henry, Referee #3**

**This manuscript presents an interesting new hypothesis explaining gravity anomalies in the Sea of Marmara area: the presence of high density bodies within the crust along the North Anatolian fault zone. However, the manuscript does not yet provide a fully convincing demonstration that the presence of these bodies is required by the available data. Owing to the non-uniqueness of gravity inversion solutions, and to the limitations of the currently available constraints from seismology, the gravity modeling alone cannot prove the existence of the high density bodies. Data may also be fit (at least at wavelengths of more than about 30 km) considering relatively small variations of Moho depth that remain compatible with constraints from seismology. The presence of high density bodies, is, however, a sound hypothesis, which can be further supported by considering the geological and geophysical contexts.**

We have carried out, as already stated above, more detailed sensitivity studies and have revised the models and interpretations (see answers to editors and reviewers #1 and #2)

**Geological knowledge on the Sea of Marmara area is already integrated in the discussion, but two important points are missing: (1) Ates et al. (1999, 2003, 2008) found magnetic anomalies in the Sea of Marmara area, which they related to the presence of magnetic bodies along the North Anatolian Fault zone. The largest one coincides with the eastern dense body infered in this study. (2) The North Anatolian fault zone follows more or less an ophiolitic suture, and this could explain at least in part the presence of dense and/or magnetic bodies along its track. Heterogeneities in the crust may thus not be a consequence of magmatic intrusions during a rifting event, but be a consequence of the convergent, and then transcurrent, tectonics during the Paleogene. This is already appearent in some of the cited references (e.g. Sengor et al., 2005) and more recent references also exist (e.g. Akbauram et al., 2016).**

Thanks for pointing us to the additional publications. We have consulted those and in particular the work on magnetic anomalies was indeed important. We have complemented the discussion with respect to these findings (see page 14-15). In particular, the we added the following paragraph to the manuscript:

Page 15 line 3: "The mechanisms and timing of the emplacement of the high-density bodies are, however, difficult to determine. The modelled density indicates that the high-density bodies represent magmatic additions to the Marmara crust, potentially originating from larger depths that rose buoyantly into domains of local extension. Magnetic anomalies across the Sea of Marmara indicate positive anomalies along the MMF that may be interpreted as magnetic bodies along the fault (Ates et al., 1999; 2003; 2008). In particular, the locations of the high-density bodies beneath the Çınarcık Basin correlate spatially with the maximum positive magnetic anomaly (Ates et al. 2008) which indicates that some mafic lithology is present there below the non-magnetic sediments."

**My conclusion would be that the gravity anomaly in the Eastern Sea of Marmara is at least in part caused by a mafic/ultramafic sliver in the crust, but it is still unclear to me whether a large high density body is present beneath Tekirdag Basin. I fully agree with the authors that these bodies could be a possible factor controling strain localization within the North Anatolian shear zone and that they predate the Pio-Quaternary transtensional tectonics, but I am not convinced they were emplaced as magmatic intrusions within the continental crust.**

Concerning this comment, we agree that the high-density bodies could also represent inherited structures. However, the spatial correlation between the position of these bodies and the thickness maxima in the syn-kinematic sediment distribution is also evident. We have therefore decided to keep the two alternative interpretation scenarios.

**Regarding the discussion with Reviewer #2, I would like to confirm that the Sandwell/TOPEX gravity model has good consistency with the marine data that were collected during Marsitecruise (both used in Kende et al., 2015), and that the Eigen- 6C4 anomaly map used here seems less consistent with these marine data. I would like to encourage the authors to go on with their suggestion to compare models fitting Topex and Eigen-6C4 gravity anomalies. I would be happy to provide the gravity data used in Kende et al. to the authors (hence, do not request to stay anonymous). Ideally, a magnetic model could be added.**

This provision of the data was essential for improving our manuscript and this way of receiving feedback is what authors ideally would wish for. As detailed in the comments to the other reviewers we have carefully carried out the comparison suggested by the reviewer and our work has greatly profited.

We agree that a magnetic model would be ideally complementing this work, but as no robust information on magnetic susceptibilities was available to us, we decided to postpone this to future work.

**References:**

Akbayram, K., Şengör, A. M. C., & Özcan, E. (2016). The evolution of the Intra-Pontide sutureâ˘Á´r: Implications of the discovery of late Cretaceous – early Tertiary mélanges. In R. Sorkhabi (Ed.), Tectonic Evolution, Collision, and Seismicity of Southwest Asia: In Honor of Manuel Berberian's Forty-Five Years of Research Contributions: Geological Society of America Special Paper 525 (Vol. 525). https://doi.org/10.1130/2016.2525(18)

Ates, A., Kayiran, T., & Sincer, I. (2003). Structural interpretation of the Marmara region, NW Turkey, from aeromagnetic, seismic and gravity data. Tectonophysics, 367, 41–99. https://doi.org/10.1016/S0040-1951(03)00044-1

Ates, A., Kearey, P., & Tufan, S. (1999). New gravity and magnetic anomaly maps of Turkey. Geophysical Journal International, 136(2), 499–502. https://doi.org/10.1046/j.1365-246X.1999.00732.x

Ates, A., Bilim, F., Buyuksarac, A., & Bektas, Ö. (2008). A tectonic interpretation of the Marmara Sea, NW Turkey from geophysical data. Earth, Planets and Space, 60(3), 169–177. https://doi.org/10.1186/BF03352780

All references were considered and integrated in our discussion.

---

## Author Comment (AC5) · 11 Mar 2019

Dear Mioara Mandea,

First of all, we would like to thank you, reviewers and all participants in the open discussion for their constructive comments and we would like to state that the format of open discussion proved to be very constructive and helpful. We particularly would like to acknowledge the help of referee Henry (reviewer #3) who not only made helpful comments but also made data available to us. He gave us access to the improved gravity data set TOPEX published by Kende et al. (2017) and to the higher resolved bathymetry of the Marmara Sea. Only thanks to this generous sharing of data we

were able to address the main concerns of reviewer #2. We would also like to thank reviewer#2 for revising his/her recommendation towards "major revisions" instead of "reject".

In our revised version we attempt to address all concerns raised by the reviewers and provide new results considering the full amount of accessible observations. Before we explain how we have addressed the specific concerns of the reviewers in detail, we would like to give a short summary of the additional work that went into the revised version of this manuscript:

We have revised the structural model by implementing the higher resolved bathymetry provided by referee Henry.

We tested the sensitivity of our results by calculating a series of "best-fit models" with respect to both gravity data sets and present a detailed discussion of these results. This quantitatively illustrates how robust the results are and in which range uncertainties are involved.

We have included the additional hypotheses concerning the origin of the modelled high-density bodies in the discussion considering the references suggested by the reviewers.

Please find our detailed answers to the reviewers' comments on AC section of the discussion forum. In summary we hope the reviewers find their concerns satisfactorily addressed and that the revised manuscript can be accepted for publication.

Yours sincerely

On behalf of all co-authors,

Ershad Gholamrezaie

---

## Author Response (AR2)

**Reply to Topical Editor Decision: Reconsider after major revisions**

Dear members of the Editorial Board,

First of all, we would like to thank you for reconsidering our manuscript for publication. Please find enclosed our detailed reply to the Reviewer #2's comments on the revised manuscript, the changes in the revised manuscript and the last version of the manuscript. In particular, we addressed the main issue of the reviewer and now explicitly indicate in the abstract and the conclusion that our solution is not unique.

We hope that all issues have been convincingly addressed and that the revised manuscript can be accepted for publication.

Yours sincerely
On behalf of all co-authors,
Ershad Gholamrezaie

**Reply to second review (Reviewer#2) of the revised manuscript**

General remark:

The authors have acknowledgedly included the available shipboard gravity data in their study. They arrive at similar conclusions as in their original version, e. g. that density contrasts are required below the Ganos and the Tuzla bends to fit the gravity field. This conclusion makes sense, as it is consistent with wide-angle reflection-refraction (WARR) seismic data suggesting abrupt uplift of crust and Moho reflectors below both bends.

WARR, however, mainly provide constraints along East-West, 2D-profiles (see Figure 3 in Bécel et al, 2009, Tectonophysics 467 , 1–21 ), but not along North-South Profiles. As a result, the 3D geometry of the deep crustal structure is very difficult to assess, based on WARR. In the Eastern Sea of Marmara, the inferred highdensity intrusion is consistent with the presence of a highly magnetized body more or less below the Tuzla bend. In contrast, in the Western Sea of Marmara, there is no evidence for any 3D, high-density crustal body below the Ganos bend, neither from multi-channel reflection seismics, neither from receiver functions along a ∼650 km transect crossing Western Anatolia at 28◦ E longitude (Karabulut et al, Geophys. J. Int, 194, 450–464). This raises questions on the reality of the inferred High sensity body below the Ganos bend. The workflow adopted by the authors is a classical one, carefully described in the paper. It consists in (1) setting up an,initial density model based on Hergert's et al previous studies ; (2) calculating the gravity response between the modelled and the observed gravity; (3) modifying the initial model by introducing additional density variations to obtain the best fitting model. The authors end up with ONE solution, which is substantially different from the picture proposed by Kende's et al (2017).

The limitations of the available seismic surveys in the Sea of Marmara is one of our motivations for the 3D gravity modelling as provides another constraint on the density distribution. The bathymetry of the Sea of Marmara especially within the North Marmara Trough (NMT) raises difficlties for seismic data processing and interpreting. In particular, the sharp scarps of the NMT cause strong diffraction that may affect the first Fresnel zone (≈ lateral resolution) during the seismic processing steps and increase the chance of data loss due to the required noise reduction. If the high-density bodies exist as dome-shape intrusive mafic bodies, as suggested by our gravity modelling, they also can diffract the seismic waves. This situation makes the high-density bodies more difficult to detect by seismic surveys. It becomes even more difficult in greater depth. Across the Sea of Marmara, most of the multi-channel reflection seismic surveys aimed to image the sediment layers and faults within the sedimentary basins. In summary, high-resolution 3D seismic survey might be an effective method to clearly image the crustal structures beneath the Princes' Islands segment and the Ganos Bend, but at the greater depth they have limits and gravity models are a complementary approach to resolve the depth distribution in particular in greater depth.

In addition, the study of Karabulut et al. (2013) aimed to image the Moho depth by the computation of receiver functions and performing common conversion point (CCP) depth migration of the P to S converted phase along the ∼650 km long NS profile in the western Anatolia. The locations of the stations in the area of the Marmara Sea do not correlate spatially with the location of the modelled thick high-density body below the Ganos Bend. Therefore, the P-wave velocity and the Vp/Vs ratio that are calculated beneath each station cannot detect the high-density body. A tomography study of Yamamoto et al. (2017), however, indicates a zone of higher S-wave velocity and slightly higher P-wave velocity at about 20 km depth b.s.l. beneath the Ganos Bend in the area where the western high-density body cuts the boundary between the upper and the lower crystalline crust.

As our work does not contribute to this topic in terms of providing new evidence based on seismic methods we leave these considerations as part of the open review-discussion and did not include them in the manuscript.

Finally, we do not agree with the reviewer that we end up with one solution. Actually, we end up with one concept that indicates lateral density heterogeneities beneath the bent segments of the Main Marmara Fault. Considering the same dataset as used in Kende et al., 2017 (Improved–TOPEX), we present two endmember solutions for the configuration of high-density bodies with different density-geometry configurations (and add more details in the supplementary information) that illustrate the spectrum of possible solutions.

However, we do agree with the reviewer that except for the tomography results of Yamamoto et al. (2017) there is no evidence for the crustal high-density body below the Ganos bend. This matter and the non-uniqueness of gravity model solutions, in general, leaves us with a question mark about the existence of the western high-density body. We discuss this in section "5.4. Model limitations" and in addition, we added the following paragraph to this section:

Page 17, Line 10: "While the aeromagnetic maps (Ates et al., 2003; 2008) indicate a clear positive anomaly (indicative for a mafic body at depth) beneath the Çınarcık Basin that spatially correlates with the eastern high-density bodies, there are no such indications for the western high-density body beneath the Ganos Bend. Considering the non-uniqueness of solutions in potential field modelling, other possible solutions based on different initial models should also be contemplated beneath the Ganos Bend (e.g. Kende et al., 2017; see Fig. S4 and S5 in the Supplement)."

Recommendation:

My recommendation is « major revisions ». The authors should insist that the solution they propose is not unique. They should discuss the possibility of fitting other solutions below the Ganos bend (e.g. Kende et al, 2017), when other initial models are choosen. The non-uniqueness issue should be explicitly mentionned in the abstract and in the conclusion.

We would like to thank the reviewer for revising his/her recommendation towards "major revisions" instead of "reject".

Regarding the discussion of "possibility of fitting other solutions below the Ganos bend", please see our answer above to the general remark.

As recommended by the reviewer, the non-uniqueness issue has been added as follows in the abstract and conclusion sections of the manuscript:

Abstract. Page 1, Line 15: "Considering the two different datasets and the general non-uniqueness in potential field modelling, we suggest three possible "endmember" solutions that are all consistent with the observed gravity field and illustrate the spectrum of possible solutions."

Conclusion. Page 19, Line 18: "(5) The configurations of the high-density bodies are exclusively based on 3D forward gravity modelling, a method characterized by an inherent non-uniqueness of the solutions. Only for the eastern bend, seismic and magnetic data support the presence of a deep high-density body, whereas for the western bend such indications are missing. Therefore, further geophysical observations are required to further constrain the detailed density-geometry configuration of these bodies."

Other remark:

Please note that in my previous review, I never claimed that the authors have used the « wrong » dataset. My concern was that the authors may have used a dataset that could not be « adapted » to the practical purpose of their study. There is a nuance between « wrong » and « not adapted ». Second, the question is not to check the quality of the « published and downloadable gravity data », but to check that the dataset they use is suitable for the objectives of their study.

We acknowledge the reviewer's concerns regarding the datasets that we had used in our first submitted manuscript and apologize for the misunderstanding related to "wrong" and "not adapted". It was a good experience to see how an open discussion could be helpful and constructive.

**3D Crustal Density Model of the Marmara Sea**

Ershad Gholamrezaie[1,2], Magdalena Scheck-Wenderoth[1,3], Judith Sippel[1], Oliver Heidbach[1], and Manfred R. Strecker[2]

[1]Helmholtz Centre Potsdam–GFZ German Research Centre for Geosciences, Potsdam, Germany
5   [2]Institute of Earth and Environmental Science, University of Potsdam, Germany
[3]Faculty of Georesources and Material Engineering, RWTH Aachen, Aachen, Germany

*Correspondence to*: Ershad Gholamrezaie (ershad@gfz-potsdam.de)

**Abstract.** The Sea of Marmara, in Northwest Turkey, is a transition zone where the dextral North Anatolian Fault Zone (NAFZ) propagates westward from the Anatolian plate to the Aegean plate. The area is of interest in the context of seismic

10   hazard in the vicinity of Istanbul, a metropolitan area with about 15 million inhabitants. Geophysical observations indicate that the crust is heterogeneous beneath the Marmara Basin, but a detailed characterization of the crustal heterogeneities is still missing. To assess if and how crustal heterogeneities are related to the NAFZ segmentation below the Marmara Sea, we develop a new crustal-scale 3D density modelmodels which integratesintegrate geological and seismological data and isthat are additionally constrained by 3D gravity modelling considering. 
[revised manuscript text omitted]